# NEURAL PREDICTOR-CORRECTOR: SOLVING HOMOTOPY PROBLEMS WITH REINFORCEMENT LEARNING

**Jiayao Mai**[1*] **Bangyan Liao**[2*] **Zhenjun Zhao**[3†] **Yingping Zeng**[1] **Haoang Li**[4]
**Javier Civera**[3] **Tailin Wu**[2] **Yi Zhou**[1✉] **Peidong Liu**[2✉]
[1]Hunan University  [2]Westlake University  [3]University of Zaragoza
[4]Hong Kong University of Science and Technology (Guangzhou)
{maijy,eeyzhou}@hnu.edu.cn,
{liaobangyan,liupeidong}@westlake.edu.cn

## ABSTRACT

The Homotopy paradigm, a general principle for solving challenging problems, appears across diverse domains such as robust optimization, global optimization, polynomial root-finding, and sampling. Practical solvers for these problems typically follow a predictor-corrector (PC) structure, but rely on hand-crafted heuristics for step sizes and iteration termination, which are often suboptimal and task-specific. To address this, we unify these problems under a single framework, which enables the design of a general neural solver. Building on this unified view, we propose **Neural Predictor-Corrector (NPC)**, which replaces hand-crafted heuristics with automatically learned policies. NPC formulates policy selection as a sequential decision-making problem and leverages reinforcement learning to automatically discover efficient strategies. To further enhance generalization, we introduce an amortized training mechanism, enabling one-time offline training for a class of problems and efficient online inference on new instances. Experiments on four representative homotopy problems demonstrate that our method generalizes effectively to unseen instances. It consistently outperforms classical and specialized baselines in efficiency while demonstrating superior stability across tasks, highlighting the value of unifying homotopy methods into a single neural framework.

## 1 INTRODUCTION

As a general principle for solving difficult problems, the Homotopy paradigm appears across diverse domains under different names, for example, Graduated Non-Convexity (Yang et al., 2020a) and Gaussian homotopy (Mobahi & Fisher III, 2015) for optimization, homotopy continuation (Bates et al., 2013) for polynomial root-finding, and annealed Langevin dynamics (Song et al., 2020) for sampling. Specifically, the Homotopy paradigm firstly construct an explicit homotopy interpolation from a simple, easily solved source problem to a complex target problem. Then, the solution of the complex problem is progressively approached by tracing the implicit trajectory along this interpolation path, effectively circumventing the challenges of direct solution.

Practical solvers for these problems often follow a predictor-corrector (PC) structure, where a predictor advances along the outer homotopy interpolation and a corrector iteratively refines the solution (Allgower & Georg, 2012). Despite their widespread use, these solvers rely on manually designed heuristics for step sizes and termination rules, which are typically suboptimal and task-specific. Furthermore, these methods have been independently developed in each domain, and no prior work has systematically unified these efforts under a single framework. We argue that this unification is crucial: it enables the design of a general solver that applies across problem instances, rather than requiring ad-hoc, per-problem solutions.

Building on this perspective, we propose **Neural Predictor-Corrector (NPC)**, a plug-and-play framework that replaces heuristic rules with automatically learned policies. Instead of manually

---

*Equal contribution.    † Project lead.    ✉Corresponding authors.

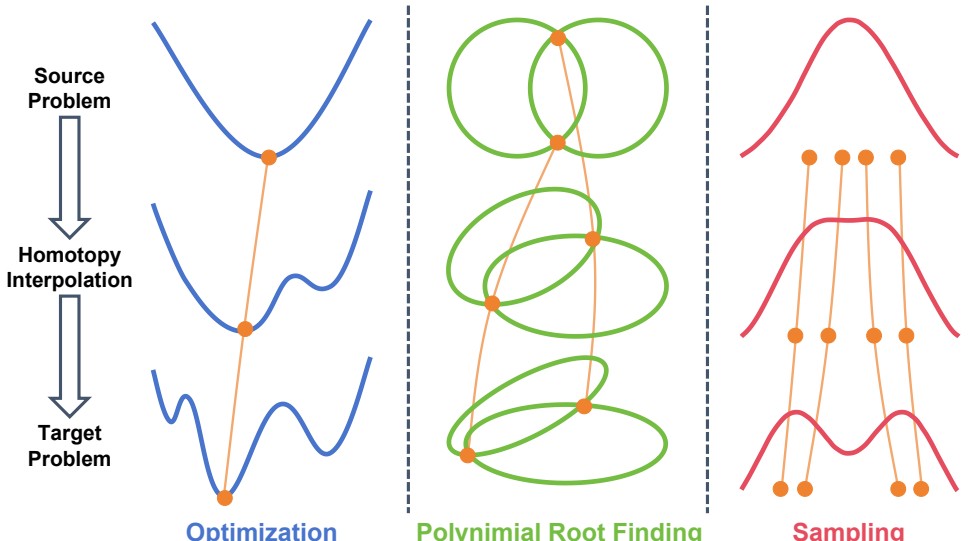

Figure 1: **Homotopy paradigm across domains.** The homotopy interpolation (blue loss functions in optimization, green polynomial roots in polynomial root-finding, and red probability densities in sampling) is explicitly defined, while the inner solution trajectory (orange curve) must be implicitly tracked.

designed rules, NPC treats the choice of predictor and corrector strategies as a sequential decision-making process (Barto et al., 1989) and employs reinforcement learning (RL) (Kaelbling et al., 1996) to adaptively learn effective policies. Crucially, we adopt an amortized training regime: a single offline training phase over a distribution of problem instances produces a policy that can be directly deployed on new instances from the same problem without per-instance fine-tuning.

We evaluate NPC on four representative homotopy tasks: Graduated Non-Convexity for robust optimization (Yang et al., 2020a), Gaussian homotopy for global optimization (Mobahi & Fisher III, 2015), homotopy continuation for polynomial root-finding (Bates et al., 2013), and annealed Langevin dynamics for sampling (Song et al., 2020). Through experiments on four representative problems, our approach is validated for strong generalization to previously unseen instances. Furthermore, the results reveal a dual advantage: our method not only consistently outperforms existing approaches in computational efficiency, but also demonstrates superior numerical stability, thereby underscoring the benefits of our proposed architecture.

In summary, our main contributions are as follows:

- To the best of our knowledge, we are the first to unify diverse problems, including robust optimization, global optimization, polynomial system root-finding, and sampling, under the homotopy paradigm, thereby revealing their common predictor-corrector structure across these problems. This enables a unified solver framework, rather than per-problem solutions.
- We introduce **Neural Predictor-Corrector (NPC)**, the first reinforcement learning-based framework that automatically learns predictor and corrector policies, replacing hand-crafted heuristics with learned, adaptive strategies.
- Extensive experiments across multiple homotopy problems demonstrate that NPC significantly outperforms other methods in efficiency, while achieving higher stability and enabling efficient, training-free deployment on previously unseen instances.

## 2 RELATED WORKS

Although PC solvers appear across multiple domains, these lines of research have largely evolved independently. We review them here and highlight gaps that motivate our work. A full discussion of related works is provided in Appendix C.

**Classical PC algorithms.** PC schemes trace solution trajectories along explicit homotopy interpolations. In robust optimization, Graduated Non-Convexity (GNC) gradually increases non-convexity to avoid poor local minima, with iterative solvers performing corrections (Yang et al., 2020a; Peng et al., 2023). Gaussian homotopy methods construct progressively less smoothed objectives to track minimizers along bandwidth reduction (Blake & Zisserman, 1987; Mobahi & Fisher III, 2015; Iwakiri et al., 2022; Xu, 2024). Polynomial system root-finding uses homotopy continuation with PC integration to trace roots from a simple start system (Bates et al., 2013; Breiding & Timme, 2018; Duff et al., 2019). In sampling, annealed Langevin dynamics and Sequential Monte Carlo define sequences of intermediate distributions with PC steps (Song & Ermon, 2019; Song et al., 2020; Doucet et al., 2001). Across all these domains, predictor and corrector components are typically hand-designed, requiring per-instance tuning and limiting generalization.

**Learning-based improvements for homotopy workflows.** Recent work has introduced learning into homotopy pipelines, showing efficient and effective improvements on Gaussian homotopy (Lin et al., 2023), sampling (Richter & Berner, 2024), combinatorial optimization (Ichikawa, 2024), and polynomial root-finding (Hruby et al., 2022; Zhang et al., 2025). However, prior learning-based methods either focus on a single homotopy component or require specialized per-instance training.

**Reinforcement learning for optimization and sampling.** RL has been applied to learn optimizers or adapt algorithmic parameters, showing benefits on some optimization and sampling tasks (Li, 2019; Belder et al., 2023; Ye et al., 2025; Liu et al., 2025; Yan et al., 2025b; Wang et al., 2025). However, these works do not address the full predictor–corrector control problem across diverse homotopy classes, nor do they leverage amortized training to produce a single policy transferable across instances.

## 3 Homotopy Paradigm as a Unified Perspective

In this section, we introduce a unified perspective on diverse problems. We begin in Sec. 3.1 by introducing the homotopy paradigm, a general principle that underlies a wide range of problems. Next, in Sec. 3.2, we show that the corresponding practical solvers can all be instantiated within a common predictor-corrector (PC) framework. Finally, in Sec. 3.3, we discuss four representative problems together with their homotopy formulations and PC implementations, thereby illustrating the breadth and utility of this unified perspective.

### 3.1 Homotopy Paradigm

As shown in Fig. 1, the homotopy paradigm provides a general principle for solving complex problem $g(\mathbf{x})$. Specifically, the homotopy paradigm defines a continuous interpolation $H(\mathbf{x}, t)$ from a simple source problem $H(\mathbf{x}, 0) = f(\mathbf{x})$ with known solutions to a complex target problem $H(\mathbf{x}, 1) = g(\mathbf{x})$. By tracing the implicit solution trajectory $\mathbf{x}^*(t)$ along this interpolation as $t$ varies from 0 to 1, one progressively transforms the source solution into the target solution. The source problem and interpolation are explicitly defined by the user, while the target solution is implicitly determined along the trajectory.

### 3.2 Predictor-Corrector Algorithm

While the homotopy paradigm specifies the abstract principle, an effective algorithm is needed to trace the implicit solution trajectory in practice. The PC method (Allgower & Georg, 2012) provides such a concrete algorithmic framework. As shown in Fig. 2, PC decomposes trajectory tracking into two complementary steps:

- **Predictor:** Determines the next level of the homotopy interpolation and predicts the solution's position at that level.
- **Corrector:** Iteratively refines the predicted solution to align it with the true solution trajectory, thereby preventing the accumulation of bias across levels.

The choice of predictor level schedule and corrector iteration count is often heuristic. Suboptimal settings can lead to inefficiency, instability, or failure to follow the trajectory accurately, motivating the development of adaptive or learning-based strategies for robust and efficient solution tracking.

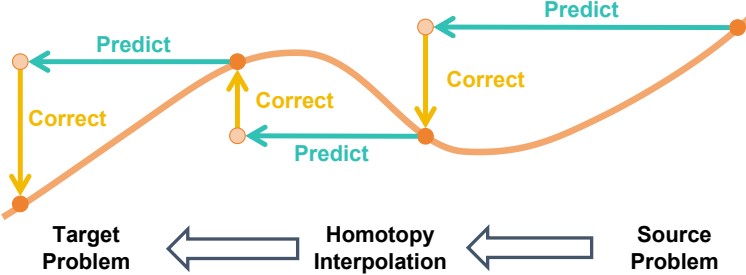

Figure 2: **Illustration of the Predictor-Corrector algorithm.** Predictor proposes the next level and provides an initial solution estimate, while Corrector iteratively refines this estimate to project it back onto the solution trajectory. Orange curve denotes the implicit solution trajectory, as in Fig. 1.

### 3.3 Representative Homotopy Problems and Practical Solvers

To illustrate the breadth of homotopy paradigm applications, we describe four representative problems together with their corresponding homotopy interpolations and PC implementations.

**1) Robust Optimization (Graduated Non-Convexity, GNC):** Robust loss functions (*e.g.*, Geman–McClure (Black & Rangarajan, 1996)) mitigate the effect of outliers. However, they introduce strong non-convexity, increasing the risk of poor local minima. Graduated Non-Convexity (GNC) (Yang et al., 2020a) addresses this challenge by defining a homotopy interpolation:

$$H(\mathbf{x}, t) = \sum_i \frac{\bar{c}^2 \, r(\mathbf{x}, y_i)^2}{\bar{c}^2 + t \, r(\mathbf{x}, y_i)^2}, \tag{1}$$

where $\bar{c}$ is a predefined parameter that controls the robustness of the GM loss, $r(\cdot, \cdot)$ represents the residual function, and $y_i$ denotes the measurements. This interpolation smoothly transitions from a convex quadratic loss ($H(\mathbf{x}, 0) = \sum_{i=1} r(\mathbf{x}, y_i)^2$) to the original non-convex Geman–McClure loss ($H(\mathbf{x}, 1) = g(\mathbf{x}) = \sum_{i=1} \frac{\bar{c}^2 \, r(\mathbf{x}, y_i)^2}{\bar{c}^2 + r(\mathbf{x}, y_i)^2}$). The predictor gradually increases non-convexity according to a predefined schedule, while the corrector refines the solution at each stage, often via a non-linear least squares optimizer (*e.g.*, Levenberg–Marquardt algorithm (Levenberg, 1944)). This homotopy strategy has proven highly effective in problems such as point cloud registration under severe outlier contamination (Yang et al., 2020b). Details are provided in Appendix A.1.

**2) Global Optimization (Gaussian Homotopy, GH):** Many optimization problems suffer from highly non-convex landscapes with narrow basins of attraction, making it difficult for solvers to converge to global or high-quality local minima. Iwakiri et al. (2022) address this challenge by progressively smoothing the target function through convolution with a Gaussian kernel $\mathcal{N}(0, t\sigma^2)$:

$$H(\mathbf{x}, t) = g(\mathbf{x}) \star \mathcal{N}(0, t\sigma^2), \tag{2}$$

where $\star$ denotes the convolution operator. This Gaussian smoothing enlarges the basin of attraction, allowing solvers to approach promising regions more reliably. The predictor progressively reduces the kernel bandwidth, while the corrector refines the solution at each stage. Details are provided in Appendix A.2.

**3) Polynomial Root-Finding (Homotopy Continuation, HC):** Root-finding for polynomial systems is challenging due to multiple solutions and computational complexity. Bates et al. (2013) address this by starting from a source system $f(\mathbf{x}) = 0$ with known roots and defining a linear homotopy:

$$H(\mathbf{x}, t) = (1 - t)f(\mathbf{x}) + tg(\mathbf{x}), \tag{3}$$

tracing the solution trajectory from the source roots to the target roots. The predictor extrapolates the next solution along this path, while the corrector refines it using Gauss-Newton (Björck, 2024) iteration at each step, ensuring accuracy along the trajectory. Details are provided in Appendix A.3.

**4) Sampling (Annealed Langevin Dynamics, ALD):** Sampling from complex, high-dimensional distributions is challenging due to multi-modality and slow mixing. Song et al. (2020) address this

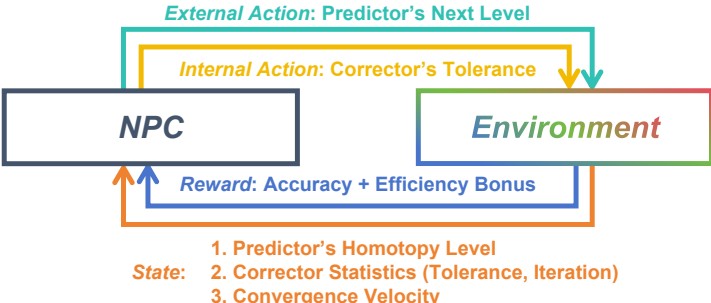

Figure 3: **RL formulation of the proposed Neural Predictor-Corrector (NPC).** At each homotopy level, the agent observes the current state (including homotopy level, corrector statistics, and convergence velocity), outputs actions that adapt the predictor's step size and the corrector's tolerance, and receives rewards designed to balance accuracy and efficiency.

by constructing a homotopy between a simple source distribution (*e.g.*, Gaussian) and the target distribution:

$$H(\mathbf{x}, t) \propto \exp\big( -(1-t)f(\mathbf{x}) - tg(\mathbf{x})\big). \tag{4}$$

The predictor schedules the intermediate distributions, while Langevin dynamics acts as the corrector at each step, iteratively refining samples to match the current intermediate distribution. Details are provided in Appendix A.4.

These examples collectively highlight the broad applicability of homotopy paradigm and the central role of predictor-corrector strategies, motivating the need for learning-based policy optimization.

## 4 NEURAL PREDICTOR-CORRECTOR WITH REINFORCEMENT LEARNING

This section introduces the Neural Predictor-Corrector (NPC) framework, a general approach for homotopy problems that replaces heuristic step-size and termination rules with neural parameterizations learned via RL. As shown in Fig. 3, NPC reformulates the predictor-corrector process as a sequential decision problem: the predictor advances the homotopy level, while the corrector ensures accuracy, both guided by adaptive policies. We first present the NPC formulation (Sec. 4.1), followed by its training with reinforcement learning (Sec. 4.2).

### 4.1 NEURAL PREDICTOR-CORRECTOR

Classical PC algorithms differ across homotopy problems in how they define prediction and correction, yet share a key limitation: their step-size schedules and termination criteria are governed by fixed heuristics. Such heuristics fail to adapt to varying solution trajectories, where small steps are needed for sharp transitions but larger steps improve efficiency when the trajectory is smooth.

The NPC addresses this limitation by parameterizing the decision rules with a neural network (NN). Instead of hand-crafted heuristics, NPC learns flexible and adaptive strategies that generalize across problem instances. The entire PC process is modeled as a Markov Decision Process (MDP), in which, at each homotopy interpolation level, an agent observes the state and selects actions that govern the procedure.

The state $s$ encodes both progress and dynamics:

---

**Algorithm 1** Neural Predictor-Corrector Solver

**Input:** Homotopy problem $H$
1: Warm up for initialization.
2: **while** $t_n \leq 1$ **do**
3:    NPC: $\{\Delta t_n, \epsilon_n \text{ or } i_n^{\max}\} = \mathbf{NN}(t_{n-1}, \epsilon_{n-1}, i_{n-1}, \tau_{n-1})$
4:    Predictor: Update interpolation level $t_n = t_{n-1} + \Delta t_n$
5:    Predictor: Predict $\mathbf{x}_{t_n}$ at level $t_n$
6:    **while** $H(\mathbf{x}_{t_n}, t_n) \leq \epsilon_n$ and $i_n \leq i_n^{\max}$ **do**
7:       Corrector: Perform one step correction
8:    **end while**
9:    Collect corrector statistics $\epsilon_n, i_n$
10:   Collect convergence velocity $\tau_n$
11: **end while**
**Output:** Optimal solution $\mathbf{x}_{t=1}^*$

---

- **Homotopy Level:** Current position along the interpolation path.

- **Corrector Statistics:** Iteration count and attained tolerance from the previous step, capturing both convergence efficiency and deviation from the predicted trajectory.
- **Convergence Velocity:** Relative change in an optimality metric between consecutive levels, reflecting the speed of convergence. For optimization and root-finding, this is the relative change in the objective value. For sampling, it is the change in a statistical distance such as Kernelized Stein Discrepancy (KSD) (Liu et al., 2016) between the empirical sample distribution and the target distribution across consecutive levels.

Given the state $s$, NPC outputs a two-part action $a$:

- **Step Size $\Delta t$:** Controls the predictor's advance along the homotopy path.
- **Corrector Termination:** Convergence threshold $\epsilon$ or maximum number of updates, balancing accuracy and efficiency.

As shown in Algorithm 1, the NPC solver operates in an iterative loop to trace the solution path of a given homotopy problem $H$. Each iteration consists of three key stages. First, a neural network (the NPC module) dynamically determines next actions for both the predictor and corrector. Second, the predictor advances the homotopy level to $t_n$ and predicts the solution $\mathbf{x}_n$ at this level. Third, the corrector iteratively refines this prediction until the convergence criteria are met. Finally, performance statistics are collected and fed back to the NPC module to inform its decisions in the next iteration, creating an adaptive, closed-loop system.

## 4.2 REINFORCEMENT LEARNING FOR NPC

Because the predictor-corrector procedure is non-differentiable and early decisions influence the entire trajectory, supervised or self-supervised training is inadequate. These approaches would require assuming that local geometric structures of the solution trajectory remain consistent across instances, which rarely holds in practice. We instead employ RL, which inherently evaluates sequential decisions by their cumulative effect and enables learning policies that generalize across instances within the same problem class. The reward function is designed to promote both accuracy and efficiency:

- **Step-wise Accuracy ($r_t^{\mathbf{acc}}$):** Encourages faithful trajectory tracking, based on convergence velocity or relative error change in the target problem.
- **Terminal Efficiency Bonus ($r^{\mathbf{eff}}$):** Rewards shorter corrector sequences, formulated as $T_{\max} - T$, where $T_{\max}$ is a predefined upper bound and $T$ is the total corrector iterations.

Consequently, the cumulative reward $R$ for an episode is defined as: $R = \left( \sum_{t=1}^{T} \lambda_1 r_t^{\mathrm{acc}} \right) + \lambda_2 r^{\mathrm{eff}}$, where $\lambda_1, \lambda_2$ are scaling coefficients detailed in Appendix A. This reward design enables agent to balance accuracy and efficiency across the homotopy trajectory.

**Remarks on amortized training for generalization.** Sequential decision-making in homotopy problems entails that early step-size choices affect all subsequent levels. Self-supervised learning fails in this context because measuring the future contribution of a step size is infeasible: it depends on the local geometric properties of the trajectory at future homotopy levels, which are unknown in advance. Relying on such assumptions risks overfitting to the training landscapes. This challenge is analogous to the dilemma discussed in (Li, 2019), where although the problem domains differ, the core issue of long-term dependency and overfitting is similar.

Reinforcement learning, by contrast, inherently evaluates actions based on cumulative outcomes, allowing NPC to adapt to diverse solution trajectories without assuming consistent local geometry. Amortized training further improves generalization: by training over a distribution of problem instances, NPC learns a policy that can be applied efficiently to unseen instances within the same problem class.

## 5 EXPERIMENTS

### 5.1 IMPLEMENTATION DETAILS

Following the RL formulation in Sec. 4, we employ Proximal Policy Optimization (PPO) (Schulman et al., 2017), an on-policy algorithm well-suited for continuous state and action spaces. Implemen-

Table 1: **Performance on the GNC point cloud registration task.** Rotation and translation errors ($E_R$ and $E_t$) are reported on a $\log_{10}$ scale.

| Sequence | Method | $\log(E_R)\downarrow$ | $\log(E_t)\downarrow$ | Iter | Time |
|---|---|---|---|---|---|
| bunny | Classic GNC | -0.85 | -2.76 | 783 | 161.00 |
| | IRLS GNC | -0.85 | -2.75 | 309 | 61.59 |
| | Ours[1]+GNC | -0.85 | -2.71 | **169** | **19.15** |
| cube | Classic GNC | -1.12 | -2.89 | 486 | 89.34 |
| | IRLS GNC | -1.10 | -2.90 | 141 | 26.13 |
| | Ours[1]+GNC | -1.11 | -2.86 | **86** | **7.86** |
| dragon | Classic GNC | -0.80 | -2.82 | 859 | 177.11 |
| | IRLS GNC | -0.80 | -2.82 | 486 | 95.93 |
| | Ours[1]+GNC | -0.80 | -2.80 | **201** | **26.42** |

[1] The agent is trained on the Aquarius sequence for the point cloud registration task.

Table 2: **Performance on the GNC multi-view triangulation task.** Reconstructed 3D point errors ($E_p$) are reported on a $\log_{10}$ scale.

| Sequence | Method | $\log(E_p)\downarrow$ | Iter | Time |
|---|---|---|---|---|
| reichstag | Classic GNC | -4.62 | 142 | 81.98 |
| | IRLS GNC | 1.74 | 37 | **10.72** |
| | Ours[1]+GNC | -4.72 | **21** | 14.18 |
| sacre_coeur | Classic GNC | -5.15 | 195 | 91.23 |
| | IRLS GNC | 0.50 | **16** | 21.31 |
| | Ours[1]+GNC | -4.84 | 20 | **14.14** |
| st_pt_sq | Classic GNC | -4.81 | 136 | 80.50 |
| | IRLS GNC | 1.00 | 19 | 27.92 |
| | Ours[1]+GNC | -4.98 | **18** | **15.55** |

[1] The agent is trained on the Aquarius sequence for the point cloud registration task.

tation is based on the open-source Stable Baselines3 library (Raffin et al., 2021). The policy and value functions are parameterized as multi-layer perceptrons (MLPs) with two hidden layers of 16 units each and ReLU activations. All other hyperparameters use the default values provided by Stable Baselines3. To account for varying problem formulations and noise levels across tasks, reward signals are scaled appropriately to ensure stable learning and comparability across tasks. Details are provided in Appendix A. All experiments are conducted on a 12-core 5.0 GHz Intel Core i7-12700KF CPU and an NVIDIA GeForce RTX 3060 GPU, unless otherwise specified.

In all tables, **Iter** ↓ records the total number of corrector iterations (rather than predictor iterations, which are more commonly used to measure progress in homotopy problems), and **Time** ↓ reports runtime in milliseconds. The best results are bolded and the second-best results in Tab. 3 are underlined. All results represent the average over 50 independent trials.

## 5.2 PROBLEM 1 : ROBUST OPTIMIZATION VIA GNC

We evaluate NPC in the context of robust optimization using the GNC framework, comparing it against the classical GNC (Classic GNC) approach and the iteratively reweighted least-squares (IRLS) version (Peng et al., 2023). The evaluation covers two spatial perception tasks with high outlier ratios: point cloud registration (Alexiou et al., 2018) (95% outliers) and multi-view triangulation (Jin et al., 2021) (50% outliers). Our NPC model is trained solely on the Aquarius dataset from the EPFL Geometric Computing Laboratory, demonstrating its cross-instance generalization capabilities.

Following the metrics defined in (Yang & Carlone, 2019), we report the rotation error ($E_R$) and translation error ($E_t$) in Tab. 1 for each method. Additionally, Tab. 2 presents the 3D point reconstruction error ($E_p$), defined as the Euclidean distance between reconstructed and ground-truth 3D points. As shown in Tabs. 1 and 2, NPC achieves accuracy comparable to Classic GNC, whereas IRLS, tailored for a specific task, performs poorly on triangulation and lacks generalization. In terms of efficiency, NPC significantly boosts GNC's performance: on point cloud registration, it reduces iterations by approximately 70-80% and runtime by 80-90% without compromising accuracy. These results demonstrate that NPC preserves the robustness of Classic GNC while substantially improving efficiency and generalization.

## 5.3 PROBLEM 2 : GLOBAL OPTIMIZATION VIA GH

We evaluate NPC in the GH setting for non-convex function minimization. We compare our method with two categories of baselines: (i) the single loop GH methods, $\text{SLGH}_r$ ($\gamma = 0.995$) and $\text{SLGH}_d$ ($\eta = 10^{-4}$) (Iwakiri et al., 2022), (ii) the Gaussian smoothing method, PGS ($N = 20$) (Xu, 2024),

and (iii) the learning-based method, CPL (Lin et al., 2023). Performance is evaluated on three 2-dimension non-convex benchmarks: the Ackley (Ackley, 1987), Himmelblau (Himmelblau et al., 1972), and Rastrigin (Rastrigin, 1974) functions. The optimal value $f(\mathbf{x}^*)$ is 0 for all problems.

As summarized in Tab. 3, NPC-accelerated GH achieves a substantial reduction in iterations and runtime compared to Classic GH, while maintaining comparable solution quality. $\text{SLGH}_d$ and PGS occasionally fail to reach the optimum, especially on Himmelblau, highlighting the challenge these landscapes pose for fixed-schedule homotopy methods. CPL is designed to learn the solution path for a specific, fixed-coefficient problem instance. Consequently, training time must be factored into the runtime, negating any efficiency advantage. Overall, these results show that NPC provides an notable trade-off between efficiency and robustness. It generalizes well to unseen problem instances while accelerating convergence.

Table 3: **Performance on GH non-convex function minimization benchmarks.**

| Problems | Method | $f(\mathbf{x}^*)\downarrow$ | Iter | Time |
|---|---|---|---|---|
| 2d Ackley | Classic GH | 0.07 | 501 | 16.25 |
| | $\text{SLGH}_r$ | 0.12 | 1839 | 56.71 |
| | $\text{SLGH}_d$ | 0.26 | 568 | 28.45 |
| | PGS | 0.07 | **200** | 14.32 |
| | CPL | 0.01 | - | 1701.61 |
| | Ours[2]+GH | 0.05 | 359 | **12.31** |
| Himmelblau | Classic GH | 0.00 | 501 | 11.39 |
| | $\text{SLGH}_r$ | 0.00 | 1839 | 41.70 |
| | $\text{SLGH}_d$ | 2.57 | **75** | **2.57** |
| | PGS | 1.18 | 200 | 11.33 |
| | CPL | 0.00 | - | 2160.17 |
| | Ours[2]+GH | 0.00 | 345 | 8.91 |
| Rastrigin | Classic GH | 0.00 | 501 | 23.76 |
| | $\text{SLGH}_r$ | 0.00 | 1839 | 78.21 |
| | $\text{SLGH}_d$ | 0.34 | 319 | 19.64 |
| | PGS | 0.14 | **200** | 11.94 |
| | CPL | 0.57 | - | 790.38 |
| | Ours[2]+GH | 0.00 | 247 | **11.84** |

[2] The agent is trained on the Ackley functions with randomized parameters and evaluated on the canonical fixed-parameter version.

## 5.4 PROBLEM 3 : POLYNOMIAL ROOT-FINDING VIA HC

We evaluate NPC in the context of polynomial system root-finding using HC. Experiments are conducted on two categories of tasks: polynomial system benchmarks (Katsura, 1990; Himmelblau et al., 1972; Rastrigin, 1974) and a computer vision problem (UPnP (Kneip et al., 2014)) for generalized camera pose estimation from 2D–3D correspondences. Tab. 4 lists the specific polynomial systems used, with the first entries as classical benchmarks and the last as computer vision task. We compare NPC-accelerated HC with Classic HC and Simulator HC (Zhang et al., 2025). Both Classic

HC and NPC-accelerated HC use the monodromy module in Macaulay2 to generate start systems following (Duff, 2021), while Simulator HC pretrains a regression neural network to predict the start system, relying on physical modeling of each problem. Consequently, Simulator HC is inapplicable to standard polynomial benchmarks. The NPC agent is trained on polynomial systems from the 4-view triangulation task with randomized coefficients to learn generalizable policies. As shown in Tab. 4, NPC consistently tracks all target solutions successfully while reducing the number of iterations and runtime compared to Classic HC. Notably, Simulator HC relies on a task-specific pre-trained network, which limits its generality, and its runtime is not directly comparable since it is implemented in C++. In contrast, NPC provides a general-purpose, adaptive solver that achieves accelerated convergence without per-task pre-training.

Table 4: **Performance on HC polynomial system benchmarks.** Succ. denotes the success rate of tracking to a root, and Time reports the average tracking time per solution path.

| Problems | Method | Succ. | Iter | Time |
|---|---|---|---|---|
| katsura10 | Classic HC | 100% | 39 | 2.22 |
| | Ours[3]+HC | 100% | **7** | **0.65** |
| cyclic7 | Classic HC | 100% | 41 | 1.96 |
| | Ours[3]+HC | 100% | **8** | **0.64** |
| UPnP | Classic HC | 100% | 53 | 8.25 |
| | Simulator HC | 100% | 100 | - |
| | Ours[3]+HC | 100% | **29** | **3.86** |

-: Runtimes are not directly comparable, as Simulator HC is implemented in C++, while the other methods are in Python.
[3] The agent is trained on polynomial systems from the 4-view triangulation task with randomized coefficients.

## 5.5 PROBLEM 4 : SAMPLING VIA ANNEALED LANGEVIN DYNAMICS (ALD)

We evaluate NPC in the context of ALD for sampling from complex distributions. Target distributions include a 40-mode Gaussian mixture model (GMM), a 10-dimensional funnel distribution (Neal, 2003), and a 4-particle double-well (DW-4) potential (Köhler et al., 2020). The NPC

agent is trained on the 10-mode GMM with randomly sampled coefficients to learn generalizable policies for accelerating ALD. We compare our method against classic ALD (Song et al., 2020) and, where applicable, iDEM (Akhound-Sadegh et al., 2024) for GMM and DW-4 with $10^3$ saved samples. Evaluation metrics are the Wasserstein-2 distance ($\mathcal{W}_2$) (Peyré et al., 2019) and the Kernelized Stein Discrepancy (KSD) (Liu et al., 2016). As shown in Tab. 5, NPC-accelerated ALD requires significantly fewer iterations while achieving $\mathcal{W}_2$ and KSD values comparable to classical ALD. Although iDEM attains lower $\mathcal{W}_2$ on some tasks, it relies on extensive per-task computation and is not directly comparable in runtime. Overall, these results demonstrate that NPC effectively accelerates sampling while maintaining high-quality approximations of the target distributions.

Table 5: **Performance on ALD sampling.** Wasserstein-2 distance ($\mathcal{W}_2$) and Kernelized Stein Discrepancy (KSD) evaluate sample quality.

| Distributions | Method | $\mathcal{W}_2 \downarrow$ | KSD $\downarrow$ | Iter | Time |
|---|---|---|---|---|---|
| 40-mode GMM | Classic ALD | 11.57 | 0.0037 | 410 | 1353.16 |
| | iDEM | 7.42 | 0.0037 | 1000 | - |
| | Ours[4]+ALD | 11.91 | 0.0040 | **110** | **772.34** |
| funnel (d=10) | Classic ALD | 30.91 | 0.0382 | 410 | 754.48 |
| | Ours[4]+ALD | 31.02 | 0.0343 | **105** | **686.55** |
| DW-4 | Classic ALD | 3.77 | 0.0911 | 410 | 1337.70 |
| | iDEM | 2.13 | 0.0911 | 1000 | - |
| | Ours[4]+ALD | 3.47 | 0.0899 | **105** | **711.66** |

-: Runtimes are not directly comparable, as iDEM is measured on a more powerful NVIDIA RTX A6000 GPU.
[4] The agent is trained on the 10-mode GMM with randomly sampled coefficients.

### 5.6 ABLATION STUDY OF RL STATE COMPONENTS

To assess the contribution of each component in the RL state, we perform an ablation study on the six datasets used for the GNC point cloud registration task, retraining the NPC agent with one component removed at a time. As summarized in Tab. 6, removing any single state component causes the agent to adopt a more conservative strategy, resulting in an increased number of corrector iterations relative to the full state. This tendency typically manifests as the agent selecting smaller predictor step sizes or stricter corrector tolerances to ensure convergence in the absence of complete information. This indicates that each state component, *i.e.*, homotopy level, corrector tolerance, corrector iteration count, and convergence velocity, provides essential information for efficiently guiding the homotopy solver. Notably, the results suggest that corrector statistics (*i.e.*, corrector tolerance and iteration) are the most informative parts of the state, as their removal leads to the largest performance drop.

Table 6: **Effect of each RL state component on NPC performance.**

| Homotopy Level | Corrector's Tolerance | Corrector's Iteration | Convergence Velocity | $\Delta$Iter |
|---|---|---|---|---|
| ✓ | ✓ | ✓ | ✓ | 0 |
| × | ✓ | ✓ | ✓ | +21 |
| ✓ | × | ✓ | ✓ | +64 |
| ✓ | ✓ | × | ✓ | +52 |
| ✓ | ✓ | ✓ | × | +38 |

### 5.7 ANALYSIS OF EFFICIENCY-PRECISION TRADE-OFF

We analyze the efficiency-precision trade-off by benchmarking our NPC-accelerated method against classical GNC and ALD. Classical approaches require manual tuning of homotopy parameters, resulting in a performance curve where higher precision typically demands more iterations. By contrast, our NPC-accelerated method bypasses this manual exploration by learning a policy that directly identifies an optimal operating point. This learned policy inherently balances the predictor step size and corrector tolerance to maximize efficiency at a given precision level. The practical benefit is visualized in Fig. 4. For both GNC and ALD tasks, the single point representing our method lies well below the classical trade-off curves, clearly illustrating a substantial reduction in iterations at comparable precision.

## 6 CONCLUSION

This paper introduces **Neural Predictor–Corrector (NPC)**, a reinforcement learning framework for homotopy solvers. By unifying diverse problems, including robust optimization, global optimization, polynomial system root-finding, and sampling, under the homotopy paradigm, their solvers

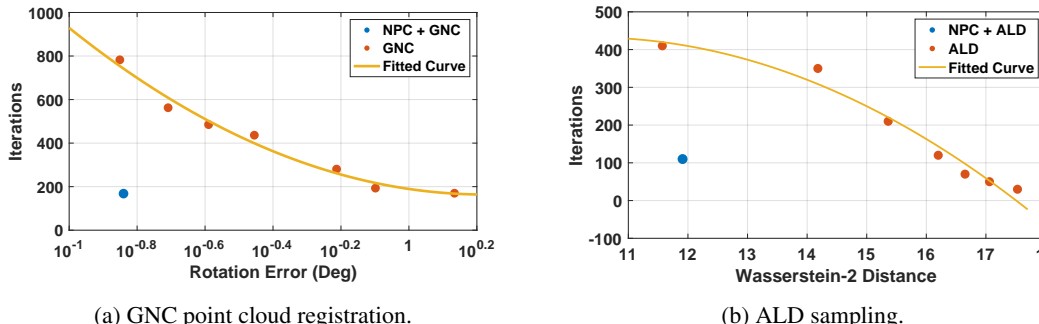

(a) GNC point cloud registration.  (b) ALD sampling.

Figure 4: **Trade-off between efficiency and precision.** Efficiency is measured in terms of corrector iterations, and precision reflects solution accuracy, for NPC-accelerated versus classical methods.

are shown to universally follow a PC structure. NPC replaces handcrafted heuristics with adaptive learned policies and employs an amortized training regime, enabling one-time offline training and efficient, training-free deployment on new instances. Extensive experiments demonstrate that NPC generalizes effectively to unseen instances, consistently outperforms existing approaches in computational efficiency, and exhibits superior numerical stability. These findings position learning-based policy search as a practical, generalizable, and efficient alternative to traditional heuristic strategies. Looking ahead, this paradigm opens promising avenues for extending homotopy methods to broader classes of optimization and sampling problems. Nonetheless, we also acknowledge its current limitation, which is discussed in Appendix D.

## 7 ACKNOWLEDGMENTS

This work was supported by the National Natural Science Foundation of China (62202389, 625B2151, 62427813, 62403401), Spanish grants PID2021-127685NB-I00 and PID2024-155886NB-I00, Aragón grant T45_23R, the AI Research and Learning Base of Urban Culture (2023WZJD008), and the Westlake University-Muyuan Joint Research Institute, the Westlake Education Foundation.

## 8 ETHICS STATEMENT

Our work unifies diverse problem domains governed by the homotopy paradigm into a single framework and, based on it, proposes a general, learning-based solver NPC. Our experiments are conducted on publicly available academic benchmarks and synthetic data, involving no human subjects or sensitive personal information. We do not foresee any direct negative societal impacts or dual-use concerns, as the primary application of our work is to provide a more efficient and robust tool for scientific inquiry.

## 9 REPRODUCIBILITY STATEMENT

To ensure reproducibility, we specify the sources for all real-world datasets and the parameters used to generate synthetic data. In addition, Appendix A provides additional implementation details, covering the specific problem formulations and the hyperparameters used in our experiments. Our code and pretrained models will also be released to the public.

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

# A    IMPLEMENTATION DETAILS

## A.1    DETAILS OF PROBLEM 1 : ROBUST OPTIMIZATION VIA GNC ( SEC. 5.2)

### A.1.1    THE GRADUATED NON-CONVEXITY ALGORITHM

Optimization problems that can be formulated as least-squares can utilize the robust kernel from Eq. (1), which is represented as:

$$\mathbf{x}^* = \min_{\mathbf{x} \in \mathcal{X}, t \in \mathcal{T}} H(\mathbf{x}, t). \tag{5}$$

The GNC algorithm utilizes Black-Rangarajan Duality (Black & Rangarajan, 1996) to reformulate Eq. (5) into:

$$\mathbf{x}^* = \min_{\mathbf{x} \in \mathcal{X}} \sum_{i=1} \left[ w_i r^2(\mathbf{y}_i, \mathbf{x}) + \Phi_{H_t}(w_i) \right], \tag{6}$$

where $w_i$ is the weight of the $i^{th}$ measurement $\mathbf{y}_i$, and the function $\Phi_{H_t}(\cdot)$, whose expression depends on the choice of the robust cost function $H_t$, defines a penalty on the weight $w_i$. When $H_t$ is defined by Eq. (1), $\Phi_{H_t}(w_i)$ is defined by $\Phi_{H_t}(w_i) = \frac{1}{t}\bar{c}^2(\sqrt{w_i} - 1)^2$. Moreover, the weight can be solved in closed form as a function of only $t$ and residual $r$.

**Predictor**    Reformulating the problem as Eq. (6) simplifies the prediction step to updating the each weight $w_i$ using Eq. (7), rather than predicting the optimization variable $\mathbf{x}$.

$$w_i = \left( \frac{\bar{c}^2}{tr^2(\mathbf{x}, \mathbf{y}_i) + \bar{c}^2} \right)^2 \tag{7}$$

**Corrector**    We correct $\mathbf{x}$ using a nonlinear optimization method defined by Eq. (8).

$$\mathbf{x}^* = \min_{\mathbf{x} \in \mathcal{X}} \sum_{i=1} w_i r^2(\mathbf{y}_i, \mathbf{x}) \tag{8}$$

In our experiments, point cloud registration employs a Gauss-Newton corrector, while multi-view triangulation uses a more robust Levenberg-Marquardt (LM) algorithm.

**Details of experiment.**    For the point cloud registration task, the reward scaling is set to $\lambda_1 = 10^3$, and $\lambda_2 = 10^{-3}$. For the multi-view triangulation task, the reward scaling is set to $\lambda_1 = 10^{-1}$ and $\lambda_2 = 10^{-3}$ due to its noise scale being significantly larger than that of point cloud registration.

## A.2    DETAILS OF PROBLEM 2 : GLOBAL OPTIMIZATION VIA GH (SEC. 5.3)

### A.2.1    THE GAUSSIAN HOMOTOPY ALGORITHM

The equivalent expression for Eq. (2) is given by:

$$H(\mathbf{x}, t) = \int g(\mathbf{x} + t * \sigma)k(\sigma)d\sigma = \mathbb{E}_{\sigma \sim \mathcal{N}(0, \mathbf{I}_d)}[g(\mathbf{x} + t * \sigma)] \tag{9}$$

where $k(\sigma) = (2\pi)^{-\frac{d}{2}} e^{\frac{-\|\sigma\|^2}{2}}$ is referred to as the kernel.

**Predictor.**    The prediction process in Gaussian homotopy is implicit, as we only modify the shape of $H(\mathbf{x}, t)$ by varying the predictor's homotopy level $t$.

**Corrector.**    The correction for $\mathbf{x}$ is performed using a momentum method (Polyak, 1964), with the gradient update formulated as

$$\begin{aligned} \mathbf{v}_{t+1} &= \nabla_{\mathbf{x}} H(\mathbf{x}_t, t) + \beta \mathbf{v}_t \\ \mathbf{x}_{t+1} &= \mathbf{x}_t - \alpha \mathbf{v}_{t+1} \end{aligned} \tag{10}$$

where $\mathbf{v}_t$ is the velocity vector, with the initial velocity $\mathbf{v}_0$ set to the zero vector, $\beta$ is momentum coefficients, controlling the influence of past gradients, and $\alpha$ is the learning rate, which determines the step size of the update. We set $\alpha = 0.01$ and $\beta = 0.8$ in our experiment. As the analytical computation of the gradient $\nabla_{\mathbf{x}} H(\mathbf{x}_t, t)$ is not feasible for some Gaussian homotopy functions, we

employ a zeroth-order method to obtain a numerical approximation. The calculation formula is as follows (Nesterov & Spokoiny, 2017):

$$\nabla_{\mathbf{x}} H(\mathbf{x}_t, t) = \nabla_{\mathbf{x}} \mathbb{E}_{\sigma \sim \mathcal{N}(0, \mathbf{I}_d)}[g(\mathbf{x} + t * \sigma)] = \frac{1}{t} \mathbb{E}_{\sigma} \left[ (g(\mathbf{x} + t * \sigma) - g(\mathbf{x})) * \sigma \right] \tag{11}$$

**Details of experiment.** The reward scaling is set to $\lambda_1 = 1$, and $\lambda_2 = 10^{-3}$.

### A.2.2 THE NON-CONVEX FUNCTION MINIMIZATION BENCHMARKS

**Ackley Optimization Problem (n-dimensions) (Ackley, 1987):**

$$f(\mathbf{x}) = -20 \exp\left(-0.2 \sqrt{\frac{1}{n} \sum_{i=1}^{n} x_i^2}\right) - \exp\left(\frac{1}{n} \sum_{i=1}^{n} \cos(2\pi x_i)\right) + 20 + e. \tag{12}$$

**Himmelblau Optimization Problem (Himmelblau et al., 1972):**

$$f(x, y) = (x^2 + y - 11)^2 + (x + y^2 - 7)^2. \tag{13}$$

**Rastrigin Optimization Problem (Rastrigin, 1974):**

$$f(x, y) = 10 + x^2 + y^2 - 9 \cos(2\pi x) - \cos(2\pi y) \tag{14}$$

### A.3 DETAILS OF PROBLEM 3 : POLYNOMIAL ROOT-FINDING VIA HC (SEC. 5.4)

### A.3.1 THE HOMOTOPY-CONTINUATION ALGORITHM

The polynomial system root-finding problem is modeled in the form of Eq. (3).

**Predictor.** The prediction of $\mathbf{x}(t + \Delta t)$ is performed using the Padé approximation. The Padé approximation polynomial $R_{n,m}(x) = \frac{R_n(x)}{Q_m(x)}$ has the following form:

$$R_{n,m}(x) = \frac{p_0 + p_1 x + \cdots + p_n x^n}{1 + q_1 x + \cdots + q_m x^m} = \frac{\sum_{j=0}^{n} p_j x^j}{1 + \sum_{k=1}^{m} q_k x^k}. \tag{15}$$

It is equivalent to the power series given in Eq. (16).

$$\psi(x) := \sum_{k=1}^{\infty} c_k x^k. \tag{16}$$

The basic Padé approximation principle is that, given two integers $m, n \in \mathbb{N} \cup \{0\}$, we can find two polynomials $P_n(x)$ of degree at most $n$ and $Q_m(x)$ of degree at most $m$, such that the difference $Q_m(x)f(x) - P_n(x)$ has an order of approximation of at least $n + m + 1$. In fact, this is mathematically equivalent to the requirement:

$$Q_m(x)f(x) - P_n(x) = O(x^{n+m+1}), \tag{17}$$

where $O(x^N)$ denotes a power series of the form $\sum_{n=N}^{\infty} c_n x^n$.

In our implementation, we set $n = 2$ and $m = 1$. we can derive the following coefficients based on Eq. (17):

$$\begin{aligned}
q_1 &= -\frac{c_3}{c_2}, \\
p_0 &= c_0, \\
p_1 &= c_1 + q_1 c_0, \\
p_2 &= c_2 + q_1 c_1,
\end{aligned} \tag{18}$$

where $c_0 = \mathbf{x}(t)$, $c_1 = \mathbf{x}'(t)$, $c_2 = \frac{1}{2}\mathbf{x}''(t)$, $c_3 = \frac{1}{6}\mathbf{x}'''(t)$. We obtain the derivative of $\mathbf{x}$ by differentiating $H(\mathbf{x}(t), t)$ with respect to t:

$$
\begin{aligned}
\frac{\partial H}{\partial \mathbf{x}}\mathbf{x}'(t) &= -\frac{\partial H}{\partial t}, \\
\frac{\partial H}{\partial \mathbf{x}}\mathbf{x}''(t) &= -\left(\frac{\partial^2 H}{\partial \mathbf{x}\partial t}\mathbf{x}'(t) + \frac{\partial^2 H}{\partial t^2}\right), \\
\frac{\partial H}{\partial \mathbf{x}}\mathbf{x}'''(t) &= -\left(2\frac{\partial^2 H}{\partial \mathbf{x}\partial t}\mathbf{x}''(t) + \frac{\partial^3 H}{\partial \mathbf{x}\partial t^2}\mathbf{x}''(t) + \frac{\partial^3 H}{\partial t^3}\right).
\end{aligned}
\tag{19}
$$

Consequently, $\mathbf{x}(t + \Delta t)$ is according to the following equation:

$$
\mathbf{x}(t + \Delta t) = \frac{p_0 + p_1\Delta t + p_2\Delta t^2}{1 + q_1\Delta t}.
\tag{20}
$$

If the denominator in Eq. (20) approaches zero, we revert to using a power series to predict $\mathbf{x}(t+\Delta t)$. In this case, the prediction takes the form $\mathbf{x}(t + \Delta t) = c_0 + c_1\Delta t + c_2\Delta t^2 + c_3\Delta t^3$.

**Corrector.** We employ a Newton corrector in our experimental setup. At each iteration, $\mathbf{x}$ is updated according to the following equation until the convergence criterion $\Delta\mathbf{x} < \epsilon$ is met.

$$
\begin{aligned}
\frac{\partial H(\mathbf{x}, t + \Delta t)}{\partial \mathbf{x}}\Delta\mathbf{x} &= -H(\mathbf{x}, t + \Delta t), \\
\mathbf{x} &= \mathbf{x} + \Delta\mathbf{x}.
\end{aligned}
\tag{21}
$$

**Details of experiment.** The reward scaling is set to $\lambda_1 = 10^{-3}$, and $\lambda_2 = 10^{-1}$.

### A.3.2 THE POLYNOMIAL SYSTEM BENCHMARKS

**The Katsura-n Polynomial System (Katsura, 1990):**

$$
\begin{aligned}
f_0 &: \quad \left(\sum_{i=-n}^{n} x_i\right) - 1 = 0 \\
f_{k+1} &: \quad x_{-n}x_n + \left(\sum_{i=-n+1}^{n} x_i x_{k-i}\right) - x_k = 0 \quad (\text{for } k = 0, 1, \dots, n-1)
\end{aligned}
\tag{22}
$$

**The Cyclic-n Polynomial System (Davenport, 1987):**

$$
\begin{aligned}
f_0 &: \quad \sum_{j=0}^{n-1} x_j = 0 \\
f_1 &: \quad \sum_{j=0}^{n-1} x_j x_{(j+1) \pmod n} = 0 \\
&\vdots \qquad\qquad \vdots \\
f_{n-2} &: \quad \sum_{j=0}^{n-1}\left(\prod_{k=0}^{n-2} x_{(j+k) \pmod n}\right) = 0 \\
f_{n-1} &: \quad \left(\prod_{j=0}^{n-1} x_j\right) - 1 = 0
\end{aligned}
\tag{23}
$$

**The Noon-n Polynomial System (Noonburg, 1989):**

$$
x_i\left(\sum_{\substack{j=1 \\ j \neq i}}^{n} x_j^2\right) - cx_i + 1 = 0 \quad \text{for } i = 1, \dots, n
\tag{24}
$$

In our implementation, we set $c = 1.1$.

**The Chandra-n Polynomial System (Kelley, 1980):**

$$2nx_k - cx_k \left( 1 + \sum_{i=1}^{n-1} \frac{k}{i+k} x_i \right) - 2n = 0 \quad \text{for } k = 1, \ldots, n \tag{25}$$

In our implementation, we set $c = 0.51234$.

### A.4 DETAILS OF PROBLEM 4 : SAMPLING VIA ALD (SEC. 5.5)

#### A.4.1 THE ANNEALED LANGEVIN DYNAMIC SAMPLING ALGORITHM

Annealed Langevin dynamics sampling obtains initial sample points from a simple distribution and uses a series of time-dependent potentials to control the update of the samples, as shown in Eq. (4). Let the time-dependent potentials be $H(\mathbf{x}, t) \propto \exp\left( -(1-t)f(\mathbf{x}) - tg(\mathbf{x}) \right)$.

**Predictor.** The prediction process in ALD sampling is implicit, as we only modify the shape of $H(\mathbf{x}, t)$ by varying the predictor's homotopy level $t$.

**Corrector.** In each iteration of the corrector, the positions of the samples are updated using the following formula:

$$\mathbf{x} = \mathbf{x} - \frac{\xi}{2} \nabla_{\mathbf{x}} H(\mathbf{x}_t, t) + \sqrt{\xi}\sigma, \tag{26}$$

where $\xi$ is a Langevin step size, and $\sigma \sim \mathcal{N}(0, \mathbf{I}_d)$ is a Gaussian noise vector.

**Details of experiment.** The reward scaling is set to $\lambda_1 = 10$, and $\lambda_2 = 10^{-3}$.

#### A.4.2 DISTRIBUTIONS

**The 10-dimensional funnel distribution.** The 10 dimensions Funnel distribution defined as

$$\begin{aligned} x_0 &\sim \mathcal{N}(0, \sigma^2) \\ x_i | x_0 &\sim \mathcal{N}(0, e^{x_0}), \quad \text{for } i = 1, \ldots, 9. \end{aligned} \tag{27}$$

The funnel potential given as

$$g(\mathbf{x}) = \frac{x_0^2}{2\sigma^2} + \frac{1}{2} \sum_{i=1}^{9} e^{-x_0} x_i^2 \tag{28}$$

**The 4-particle double-well (DW-4) potential.** The DW-4 potential defined as

$$g(\mathbf{x}) = \frac{1}{2\tau} \sum_{ij} a(d_{ij} - d_0) + b(d_{ij} - d_0)^2 + c(d_{ij} - d_0)^4, \tag{29}$$

where $d_{ij} = \|x_i - x_j\|_2$ is the Euclidean distance between particles $i$ and $j$. In our implementation, we set $\tau = 1$, $a = 0$, $b = -4$, and $c = 0.9$.

#### A.4.3 METRIC

**Wasserstein-2 distance ($\mathcal{W}_2$).** The Wasserstein-2 distance (Peyré et al., 2019) is given by

$$\mathcal{W}_2(\mu, \nu) = \left( \inf_{\pi} \int \pi(x, y) d(x, y)^2 \, dx dy \right)^{\frac{1}{2}}, \tag{30}$$

where $\pi$ is the transport plan with marginals constrained to $\mu$ and $\nu$ respectively. In our implementation, we we use the Python Optimal Transport (POT) package (Flamary et al., 2021) to compute this metric.

**Kernelized Stein Discrepancy (KSD).** The Kernelized Stein Discrepancy (Liu et al., 2016) is defined as

$$u_q(x, x') = s_q(x)^\top k(x, x') s_q(x') + s_q(x)^\top \nabla_{x'} k(x, x')$$
$$+ \nabla_x k(x, x')^\top s_q(x') + \text{trace}(\nabla_{x,x'} k(x, x')), \tag{31}$$
$$\mathbb{S}(p, q) = \mathbb{E}_{x,x'\sim p}[u_q(x, x')],$$

where $s_q = -\nabla_{\mathbf{x}} g(\mathbf{x})$, and $k(x, x')$ is a positive definite kernel. Specifically, we use the standard RBF kernel for KSD computation in this work.

## B  BACKGROUND ON REINFORCEMENT LEARNING

Reinforcement learning (RL) (Kaelbling et al., 1996) provides a natural framework for learning adaptive strategies. It formulates sequential decision-making as an Markov Decision Process (MDP) with state space $\mathcal{S}$, action space $\mathcal{A}$, transition dynamics $p(s_{t+1}|s_t, a_t)$, initial state distribution $p_0(s_0)$, reward function $r(s_t, a_t)$, and discount factor $\gamma \in (0, 1]$. The goal is to find an optimal policy $\pi^* : \mathcal{S} \to \mathcal{A}$ that maximizes the expected cumulative reward along a trajectory $(s_0, a_0, \ldots, s_T)$:

$$\mathbb{E}\left[\sum_{t=0}^{T} \gamma^t r(s_t, a_t)\right]. \tag{32}$$

## C  FULL RELATED WORK

Although we mentioned in previous sections that methods from different fields essentially share the same predictor-corrector spirit, they have long evolved independently of each other. Our work is the first to unify these methods. In this section, we will review 1) classical predictor-corrector methods; 2) learning-based improvements on predictor-corrector methods; 3) efficient optimization and sampling methods via reinforcement learning.

**Classical PC algorithms.**  **1)** Robust optimization: A core related technique is Graduated Non-Convexity (GNC), first proposed by (Yang et al., 2020a). This method employs a predictor-corrector approach with non-linear least-squares solvers to compute robust solutions. However, it relies on a hand-crafted, fixed iteration schedule, making it unsuitable for real-time robotics applications. Building on this work, Peng et al. (2023) established a connection between GNC and the iteratively reweighted least-squares (IRLS) framework, based on which they designed a novel iteration strategy that achieved faster speeds in point cloud registration tasks (Liu et al., 2023; Yan et al., 2025a; Chen et al., 2025a; Liao et al., 2024). Nevertheless, this strategy's lack of generalizability to other problems remains its primary limitation. **2)** Gaussian homotopy optimization: The underlying principle of this area was first introduced in (Blake & Zisserman, 1987). More recently, Iwakiri et al. (2022) proposed a novel single-loop framework for the Gaussian homotopy method that simultaneously performs prediction and correction. Subsequently, Xu (2024) improved the algorithm's convergence rate by adding an exponential power-N transformation prior to the Gaussian homotopy process. **3)** Polynomial root-finding: Homotopy continuation (Bates et al., 2013), a numerical method for finding the roots of polynomial systems, uses a predictor-corrector scheme to track solution paths. Subsequent methods by (Breiding & Timme, 2018) and (Duff et al., 2019) analyzed the properties of polynomials to introduce various improvements, enhancing the algorithm's speed. **4)** Sampling: In generative modeling, annealed Langevin dynamics (Song & Ermon, 2019; Song et al., 2020) utilizes a predictor-corrector method to sample from image probability distributions, where the correction step uses Langevin dynamics to restore samples to an equilibrium state. Similarly, Sequential Monte Carlo (SMC) methods (Doucet et al., 2001) also apply a predictor-corrector approach to sample from posterior probability distributions, with a correction step that employs importance sampling to re-weight the samples.

**Learning-based improvements for homotopy workflows.**  **1)** Gaussian homotopy optimization: Lin et al. (2023) is a novel model-based method that learns the entire continuation path for Gaussian homotopy. However, this approach requires specialized training for each problem. **2)** Polynomial root-finding: Focusing on the more specific sub-problem of polynomial system root-finding, both (Hruby et al., 2022) and (Zhang et al., 2025) propose learning-based methods to determine

the optimal starting system for homotopy continuation. **3) Combinatorial optimization:** Ichikawa (2024) proposes the Continuous Relaxation Annealing strategy, aiming to enhance unsupervised learning solvers for combinatorial optimization problems. **4) Sampling:** Richter & Berner (2024) establishes a unifying framework based on path space measures and time-reversals, and proposes a novel log-variance loss that avoids differentiation through the SDE solver.

**Reinforcement learning for optimization and sampling.** **1)** Optimization: Li (2019) proposes a general framework by formulating an optimization algorithm as a reinforcement learning problem, where the optimizer is represented as a policy that learns to generate update steps directly, aiming to converge faster and find better optima than hand-engineered method. Belder et al. (2023) utilizes reinforcement learning (Chen et al., 2025b) to train an agent that dynamically selects the damping factor in the Levenberg-Marquardt algorithm (Levenberg, 1944) to accelerate convergence by reducing the number of iterations. **2)** Sampling: Ye et al. (2025) employs reinforcement learning to adaptively predict the denoising schedule via optimizing a reward function that encourages high image quality while penalizing an excessive number of denoising steps. Wang et al. (2025) proposes a general framework named Reinforcement Learning Metropolis-Hastings, which aims to automatically design and optimize Markov Chain Monte Carlo (MCMC) samplers.

## D  LIMITATION AND FUTURE WORK

One limitation of our work is that the NPC agent's reward scale currently requires manual tuning for each problem instance based on its noise level to ensure stable and efficient training. The scale of step-wise rewards influences the training process's convergence time, while an oversized terminal reward can nullify the guidance from step-wise rewards. This imbalance can prevent the agent from correctly tracking the solution trajectory, causing it to adopt myopic strategies to prematurely reach a terminal state. We conduct experiments on the point cloud registration task with different reward scaling factors. The results are shown in Tab. 7.

Table 7: **Comparison of results under different reward scaling settings.** Convergence Steps (Training) denotes the approximate step count where the cumulative reward stabilizes during training.

| Method | Reward Scaling | Convergence Steps (Training) | $\log(E_R)\downarrow$ | $\log(E_t)\downarrow$ | Iter |
|---|---|---|---|---|---|
| Ours+GNC | $\lambda_1 = 10^3, \lambda_2 = 10^{-3}$ (*) | 3M | -1.11 | -2.86 | 86 |
| | $\lambda_1 = 10^2, \lambda_2 = 10^{-3}$ | 2M | -1.08 | -2.67 | 70 |
| | $\lambda_1 = 10^3, \lambda_2 = 10^{-4}$ | 6M | -1.08 | -2.91 | 74 |
| | $\lambda_1 = 10^2, \lambda_2 = 10^{-2}$ | Fail | - | - | - |
| Classic GNC | - | - | -1.12 | -2.89 | 486 |
| IRLS GNC | - | - | -1.10 | -2.90 | 141 |

(*): The settings used in the paper.

To address this, two avenues for future work are promising. The first is to develop a mechanism that automatically adapts the reward scale. A more fundamental solution would be to investigate adaptive normalization techniques for the reward function, making the learning process inherently robust and eliminating manual tuning.

## E  FULL EXPERIMENTAL RESULTS

This section provides complete experimental results, which are summarized in Secs. 5.2 to 5.4 due to limited space. Tab. 8 shows the full results for the point cloud registration experiments via GNC, Tab. 9 shows the full results for the non-convex function minimization experiments via GH, and Tab. 10 presents the detailed results for the root-finding experiments on polynomial systems via HC. In addition, we present box plots for a subset of the experimental results in Fig. 5 to visually compare the different methods.

Table 8: **Performance on the GNC point cloud registration task.** Rotation and translation errors ($E_R$ and $E_t$) are reported on a $\log_{10}$ scale.

| Sequence | Method | $\log(E_R)\downarrow$ | $\log(E_t)\downarrow$ | Iter | Time |
|---|---|---|---|---|---|
| bunny | Classic GNC | -0.85 | -2.76 | 783 | 161.00 |
| | IRLS GNC | -0.85 | -2.75 | 309 | 61.59 |
| | Ours[1]+GNC | -0.85 | -2.71 | **169** | **19.15** |
| cube | Classic GNC | -1.12 | -2.89 | 486 | 89.34 |
| | IRLS GNC | -1.10 | -2.90 | 141 | 26.13 |
| | Ours[1]+GNC | -1.11 | -2.86 | **86** | **7.86** |
| dragon | Classic GNC | -0.80 | -2.82 | 859 | 177.11 |
| | IRLS GNC | -0.80 | -2.82 | 486 | 95.93 |
| | Ours[1]+GNC | -0.80 | -2.80 | **201** | **26.42** |
| egyptian_mask | Classic GNC | -0.88 | -2.73 | 770 | 160.05 |
| | IRLS GNC | -0.86 | -2.75 | 264 | 53.51 |
| | Ours[1]+GNC | -0.87 | -2.69 | **158** | **16.94** |
| sphere | Classic GNC | -0.98 | -2.87 | 713 | 148.55 |
| | IRLS GNC | -0.98 | -2.88 | 220 | 45.73 |
| | Ours[1]+GNC | -0.99 | -2.77 | **143** | **13.63** |
| vase | Classic GNC | -0.86 | -2.84 | 765 | 159.25 |
| | IRLS GNC | -0.87 | -2.86 | 288 | 58.08 |
| | Ours[1]+GNC | -0.86 | -2.77 | **160** | **17.05** |

[1] The agent is trained on the Aquarius sequence for the point cloud registration task.

Table 9: **Performance on GH non-convex function minimization benchmarks.**

| Problems | Method | $f(\mathbf{x}^*)\downarrow$ | Iter | Time |
|---|---|---|---|---|
| 2d Ackley | Classic GH | 0.07 | 501 | 16.25 |
| | SLGH$_r$ | 0.12 | 1839 | 56.71 |
| | SLGH$_d$ | 0.26 | 568 | 28.45 |
| | PGS | 0.07 | **200** | 14.32 |
| | CPL | 0.01 | - | 1701.61 |
| | Ours[2]+GH | 0.05 | 359 | **12.31** |
| Himmelblau | Classic GH | 0.00 | 501 | 11.39 |
| | SLGH$_r$ | 0.00 | 1839 | 41.70 |
| | SLGH$_d$ | 2.57 | **75** | **2.57** |
| | PGS | 1.18 | 200 | 11.33 |
| | CPL | 0.00 | - | 2160.17 |
| | Ours[2]+GH | 0.00 | 345 | 8.91 |
| Rastrigin | Classic GH | 0.00 | 501 | 23.76 |
| | SLGH$_r$ | 0.00 | 1839 | 78.21 |
| | SLGH$_d$ | 0.34 | 319 | 19.64 |
| | PGS | 0.14 | **200** | 11.94 |
| | CPL | 0.57 | - | 790.38 |
| | Ours[2]+GH | 0.00 | 247 | **11.84** |
| 10d Ackley | Classic GH | 0.01 | 501 | 27.58 |
| | SLGH$_r$ | 0.02 | 1839 | 91.90 |
| | SLGH$_d$ | 0.37 | 435 | 33.58 |
| | Ours[2]+GH | 0.47 | **398** | **10.88** |

[2] The agent is trained on the Ackley functions with randomized parameters and evaluated on the canonical fixed-parameter version.

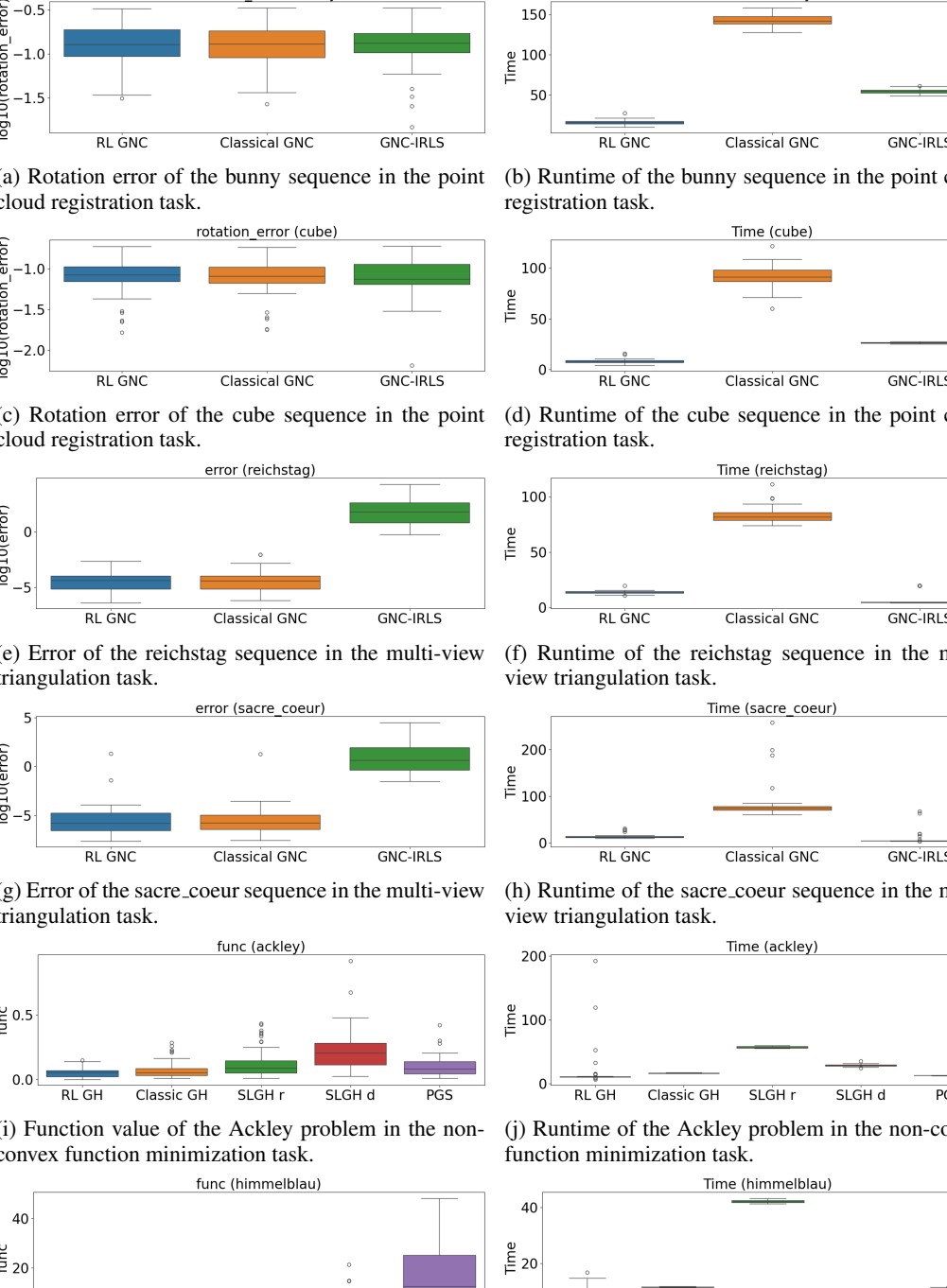

(a) Rotation error of the bunny sequence in the point cloud registration task.

(b) Runtime of the bunny sequence in the point cloud registration task.

(c) Rotation error of the cube sequence in the point cloud registration task.

(d) Runtime of the cube sequence in the point cloud registration task.

(e) Error of the reichstag sequence in the multi-view triangulation task.

(f) Runtime of the reichstag sequence in the multi-view triangulation task.

(g) Error of the sacre_coeur sequence in the multi-view triangulation task.

(h) Runtime of the sacre_coeur sequence in the multi-view triangulation task.

(i) Function value of the Ackley problem in the non-convex function minimization task.

(j) Runtime of the Ackley problem in the non-convex function minimization task.

(k) Function value of the Himmelblau problem in the non-convex function minimization task.

(l) Runtime of the Himmelblau problem in the non-convex function minimization task.

Figure 5: **Supplementary box plots of performance metrics.** These visualizations illustrate the result distributions over 50 independent trials, providing a more intuitive understanding of the stability and efficiency of each method.

Table 10: **Performance on HC polynomial system benchmarks.** Succ. denotes the success rate of tracking to a root, and Time reports the average tracking time per solution path.

| Problems | Method | Succ. | Iter | Time |
|---|---|---|---|---|
| katsura10 | Classic HC | 100% | 39 | 2.22 |
| | Ours[3]+HC | 100% | **7** | **0.65** |
| cyclic7 | Classic HC | 100% | 41 | 1.96 |
| | Ours[3]+HC | 100% | **8** | **0.64** |
| noon5 | Classic HC | 100% | 41 | 1.69 |
| | Ours[3]+HC | 100% | **10** | **0.69** |
| chandra9 | Classic HC | 100% | 31 | 3.24 |
| | Ours[3]+HC | 100% | **5** | **0.76** |
| UPnP | Classic HC | 100% | 53 | 8.25 |
| | Simulator HC | 100% | 100 | - |
| | Ours[3]+HC | 100% | **29** | **3.86** |

-: Runtimes are not directly comparable, as Simulator HC is implemented in C++, while the other methods are in Python.
[3] The agent is trained on a separate set of polynomial systems with randomized coefficients.

## F THE USE OF LARGE LANGUAGE MODELS (LLMS)

We use LLMs to polish writing.

