# OpenReview forum: "Neural Predictor-Corrector: Solving Homotopy Problems with Reinforcement Learning"
_ICLR.cc/2026/Conference — ICLR 2026 Poster_

### Official Review · Reviewer_CRzY · 2025-10-23

**Soundness:** 3
**Presentation:** 3
**Contribution:** 3
**Rating:** 6
**Confidence:** 3

**Summary:**

A number of problems in different domains can be framed as homotopy
interpolations between an easily solved source problem and a complex target
problem. Such problems are typically solved with a predictor-corrector
framework. Here, a solution is found to the initial easy problem. Then, the
predictor advances the problem following the homotopy interpolation, and the
corrector modifies the solution to fit the new problem.

Prior work has treated these homotopy problems separately in different domains.
This paper proposes to unite a number of these problems under a single
framework. It then proposes a neural predictor-corrector framework, which uses
reinforcement learning to train a policy that outputs the next level for the
predictor and the next tolerance for the corrector --- previously, these values
were determined via heuristics. Experiments show that the final RL agent
effectively learns to set these values.

**Strengths:**

1. This application of reinforcement learning seems quite novel to me. I
   appreciate that PPO is able to be used "out of the box" to solve the problem.
2. The problems chosen for the paper are quite diverse, which shows the
   applicability of the framework. For example, I could see the application to
   sampling / Langevin dynamics could eventually tie into generative modeling.
3. The computational requirements for this method are quite low (lines 314-315),
   making the research more accessible.
4. The limitation of sensitivity to reward scale is acknowledged in the
   appendix.
5. The ablation of the state components helps in showing that all parts of the
   state are necessary for NPC.

Overall, I found the paper quite easy to read despite not having much
familiarity with homotopy interpolations myself.

**Weaknesses:**

My main concern is that it seems the experiments were only conducted over one
trial, as I could not find any mention of repetitions, and no error bars are
reported in the tables. It is advisable to conduct multiple trials of each
algorithm and use statistical testing to check that differences between
algorithms are significant.

I have also listed several questions in the section below; these are minor points that I think would be useful to address in the updated paper.

**Questions:**

1. I am unclear if the Homotopy Paradigm discussed in Sec. 3.1 is a novel
   contribution of this work. Is the term "Homotopy Paradigm" already widely
   used in the literature? If so, could a citation for it be added in Sec. 3.1?
   I think I am confused because the first sentence of the introduction
   (line 32) seems to indicate that the Homotopy paradigm is already well-known,
   while lines 121-122 indicate that it is a perspective introduced in this
   work. Assuming this perspective is novel, it may be good to list it as a
   contribution in the introduction. Right now, the first contribution of the
   introduction (line 82) makes it clear that unifying the approaches under the
   homotopy paradigm is novel, but it is unclear if the homotopy paradigm itself
   is novel.
2. Based on Appendix D, it seems the NPC framework shifts the manual effort from
   tuning the parameters for the predictor and corrector to tuning the reward
   scales for the RL training. What do you see as the benefits and tradeoffs of
   this shift? Might there be cases where it is easier to just use the
   heuristics? When is it better to try to tune the rewards?
3. Table 6 shows that removing certain components from the state can increase
   the number of iterations required more than removing other components, e.g.,
   removing the corrector's tolerance increases by 64, while removing the
   homotopy level only increases by 21. Does this say anything about which parts
   of the state are most essential to NPC?

---

> ### Author Response · Authors · 2025-11-19
> **Official Response by Authors**
>
> We thank the reviewer for the constructive comments. We are pleased to provide the following clarifications regarding the points raised.
>
> > **W1: My main concern is that it seems the experiments were only conducted over one trial, as I could not find any mention of repetitions, and no error bars are reported in the tables. It is advisable to conduct multiple trials of each algorithm and use statistical testing to check that differences between algorithms are significant.**
>
> All reported results represent the average of 50 independent trials. We report only mean values in the main tables due to space constraints. We have clarified this in the revised manuscript and provided supplementary box plots in the Appendix E (Line 318).
>
> > **Q1: I am unclear if the Homotopy Paradigm discussed in Sec. 3.1 is a novel contribution of this work. Is the term "Homotopy Paradigm" already widely used in the literature? Assuming this perspective is novel, it may be good to list it as a contribution in the introduction. Right now, the first contribution of the introduction (line 82) makes it clear that unifying the approaches under the homotopy paradigm is novel, but it is unclear if the homotopy paradigm itself is novel.**
>
> We thank the reviewer for pointing out the ambiguity.
> The term “Homotopy Paradigm” is not an established term in the existing literature and is introduced in our work as a unifying perspective.
> We have revised the introduction (Line 32-39) to list the term "Homotopy Paradigm" as a perspective introduced by us.
>
> > **Q2: Based on Appendix D, it seems the NPC framework shifts the manual effort from tuning the parameters for the predictor and corrector to tuning the reward scales for the RL training. What do you see as the benefits and tradeoffs of this shift? Might there be cases where it is easier to just use the heuristics? When is it better to try to tune the rewards?**
>
> Similar concern as the Weaknesse 3 from Reviewer 7Gmc. We have reproduced our response below for your convenience.
>
> Although the NPC method requires manual reward scaling, it is not sensitive to the specific scale parameters, as shown in the following table. Specifically, we generally determine the reward scale by roughly matching the orders of magnitude of the two reward components. Once determined, this scale remains fixed for all following experiments.
>
> | Method | Reward Scaling | Convergence Steps (Training) | $\log(E_R) \downarrow$ | $\log(E_t) \downarrow$ | Iter |
> | :--- | :--- | :---: | :---: | :---: | :---: |
> | Ours+GNC | $\lambda_1 = 10^3, \lambda_2 = 10^{-3} (*)$ | 3M | -1.11 | -2.86 | 86 |
> | | $\lambda_1 = 10^2, \lambda_2 = 10^{-3}$ | 2M | -1.08 | -2.67 | 70 |
> | | $\lambda_1 = 10^3, \lambda_2 = 10^{-4}$ | 6M | -1.08 | -2.91 | 74 |
> | | $\lambda_1 = 10^2, \lambda_2 = 10^{-2}$ | Fail | - | - | - |
> | Classic GNC | - | - | -1.12 | -2.89 | 486 |
> | IRLS GNC | - | - | -1.10 | -2.90 | 141 |
>
> > $(*)$: The settings used in the paper.
>
> > **Q3: Does this say anything about which parts of the state are most essential to NPC?**
>
> Table 6 indicates that the corrector statistics (i.e., corrector tolerance and iteration) are the most informative parts of the state, as their removal leads to the largest performance drop. We have added this clarification to the ablation study in the revised manuscript (Line 441-445).

---

> > ### Comment · Reviewer_CRzY · 2025-11-21
> >
> > I thank the authors for taking the time to address my comments. My main concern over the rigor of the experiments has been addressed, as well as my minor questions. I appreciate the inclusion of the box plots and the clarification regarding the homotopy paradigm. As such, I have raised my score to an accept.

---

> > > ### Author Response · Authors · 2025-11-23
> > > **Official Comment by Authors**
> > >
> > > Thank you for your response.
> > > We are delighted to hear that our reply addressed your concerns and appreciate your recognition of our work. We sincerely appreciate your thorough and valuable feedback that has been helpful to our work.

---

### Official Review · Reviewer_pEeU · 2025-10-23

**Soundness:** 3
**Presentation:** 3
**Contribution:** 3
**Rating:** 6
**Confidence:** 3

**Summary:**

The homotopy paradigm is a principle to solve optimization problems by transporting solutions to problems from a simple family to a target problem domain. The paper discusses application areas in which this principle is applied: annealing in optimization and sampling. The predictor corrector (PC) method is a family of methods that solve homotopy problems by predicting a new location of a solution and then correcting that prediction. Existing PC approaches typically rely on a an interpolation schedule that must be chosen in advance. While some domains offer natural choices, others do not as the authors claim. Therefore, the authors propose to learn an interpolation using RL, which is implemented by learning the predictor and corrector steps using a neural network. The evaluation of the paper contains tasks from 4 domains and ablation studies that showcase the generalizability of the approach and competitive performance. The ablations provide some insights into the importance of the individual components.

**Strengths:**

- The paper provides a nice unifying perspective that was at least new to me.
- The method seems to improve over baselines on the reported problems. For some of the domains they evaluate on (GNC, root finding) I am not sure how well chosen the baselines are but that is rather on me.
- The method is pretty simple, and thus seems to be easy to reimplement. I am surprised that almost no hyperparameters must be changed from the stablebaselines defaults.
- Overall the paper is pretty clear and well written.

**Weaknesses:**

### Method
- The paper presents a very interesting and simple idea: use (reinforcement) learning to improve optimization and sampling methods. While I am not aware of any paper that discusses this under a single umbrella of homotopy problems, I have seen works that use learning to propose sampling steps, eg [1, 2, 3, 4, 5]. Some of these methods are mentioned in the related work section, yet I think it should become more clear that the objective of the paper is a unifying perspective.

### Wording:
- "Because the predictor-corrector procedure is non-differentiable and early decisions influence the entire trajectory, supervised or self-supervised training is inadequate" I think its totally fair that you use RL, but if you had data for supervised training, I think you could actually train in supervised fashion as your NN will be differentiable, no?
- I am not sure if you can call the predictor and corrector schedules in Song et al. (2020) "handcrafted heuristics " as you state in your text, but their design choices are theoretically well motivated. Generally, using a linear schedule makes sense for many problems I would say.

### Evaluation
- The global optimzation problems are only in 2d, which seems pretty low to me, given that other communities like the derivative-free optimization community optimize on those functions in >10d. I would be interested in seeing the results on higher dimensions. But I think this point alone is not enough for rejection in my eyes.
- I think the results are not the strongest on every task, for instance on the sampling problems, PGS seems to be on par at least. But I think this is a nitpick, since the approach seems to be more motivated to convince in its generality.

### Clarity
- While the high level idea of the paper is clear to me, some details of the algorithm are not: Why do you predict the corrector actions in line 3 of the algorithm? Is it always that single action you apply in line 7? If you predict both steps at once why have them separate at all? It would make much more sense to me if you first predicted the predictor step and then iteratively multiple different corrector steps based on H.

### Minor
- You should mention in the main paper that the functions you test on are 2d in Section 5.3.

---
### Sources
[1] Richter, Lorenz, and Julius Berner. "Improved sampling via learned diffusions." _arXiv preprint arXiv:2307.01198_ (2023).

[2] Wang, Congye, et al. "Reinforcement learning for adaptive MCMC." _arXiv preprint arXiv:2405.13574_ (2024).

[3] Xi Lin, Zhiyuan Yang, Xiaoyuan Zhang, and Qingfu Zhang. Continuation path learning for homotopy optimization. In International Conference on Machine Learning, pp. 21288–21311. PMLR, 2023.

[4] Ichikawa, Yuma. "Controlling continuous relaxation for combinatorial optimization." _Advances in Neural Information Processing Systems_ 37 (2024): 47189-47216.

[5] Hruby, Petr, et al. "Learning to solve hard minimal problems." _Proceedings of the IEEE/CVF Conference on Computer Vision and Pattern Recognition_. 2022.

**Questions:**

- How do you weigh the two terms in your reward formulation? A specific equation would be very helpful.
- For how long did you train your model? I am not familiar with the defaults of stable baselines and the info is not listed in the appendix.
- How did you choose your kernel for computing the KSD in the evaluation?
- As you state yourself, there are prior methods that use learning in the context of PC, but which seem to not generalize as well as yours as you point out in line 111. It would be interesting to see a comparison here, to understand better at which cost the generalization of your method comes, if an at all, especially as it is one of the key claimed contributions. Have you done such comparisons?

---

> ### Author Response · Authors · 2025-11-19
> **Official Response by Authors -- Part 1**
>
> We greatly appreciate the insightful comments. We have carefully reviewed them and would like to address the reviewer's concerns as follows.
>
> > **W1: Some of these methods are mentioned in the related work section, yet I think it should become more clear that the objective of the paper is a unifying perspective.**
>
> We thank the reviewer for highlighting this point.
> The unifying perspective is indeed the central contribution of our work. Actually, we have emphasized this perspective throughout the paper as highlighted in Line 16-17, Line 45-47, and Line 82.
> In the revised manuscript, we have added the relevant references [1–5] mentioned by the reviewer (Line 110-111, 917-921, 930-932).
>
> > **W2: I think its totally fair that you use RL, but if you had data for supervised training, I think you could actually train in supervised fashion as your NN will be differentiable, no?**
>
> While the neural network outputs a schedule, the overall predictor–corrector procedure remains non-differentiable due to the non-linear optimization inside each corrector step.
> This non-differentiability makes supervised training impractical, and even if applied, it would likely yield a sub-optimal policy by imitating heuristics.
> In contrast, reinforcement learning is naturally suited to discover optimal policies in such complex, non-differentiable settings.
>
> > **W3: I am not sure if you can call the predictor and corrector schedules in Song et al. (2020) "handcrafted heuristics " as you state in your text, but their design choices are theoretically well motivated. Generally, using a linear schedule makes sense for many problems I would say.**
>
> We classify the schedule in Song et al. (2020)[1] as a handcrafted heuristic.
> While such linear schedules are reasonable baselines, they remain sub-optimal.
> Our work, in contrast, utilizes reinforcement learning to discover the optimal schedule.
>
> > **W4: I would be interested in seeing the results on higher dimensions.**
>
> The selection of global optimization tasks follows standard benchmarks used in prior Gaussian Homotopy literature[2,3]. Nevertheless, to address the reviewer's interest in higher-dimensional settings, we conduct an additional evaluation on the 10-dimensional Ackley function. The results are presented in the table below.
>
> | Problems | Method | $f(\mathrm{x}^*) \downarrow$ | Iter | Time |
> | :--- | :--- | :---: | :---: | :---: |
> | 10d Ackley | Classic GH | 0.01 | 501 | 27.58 |
> | | SLGH$_r$ | 0.02 | 1839 | 91.90 |
> | | SLGH$_d$ | 0.37 | 435 | 33.58 |
> | | Ours$^2$+GH | 0.47 | 398 | 10.88 |
>
> > **W5: I think the results are not the strongest on every task, for instance on the sampling problems, PGS seems to be on par at least.**
>
> While PGS demonstrates competitive efficiency in Table 3, it notably falls into a local optimum on the Himmelblau task. In contrast, our method consistently ranks within the top two in efficiency and exhibits superior stability, successfully converging to the global minimum across all instances. Additionally, we have included a box plot from the revised manuscript (page 20), where visually demonstrates the stability of our method.
>
> > **W6: Why do you predict the corrector actions in line 3 of the algorithm? Is it always that single action you apply in line 7? If you predict both steps at once why have them separate at all? It would make much more sense to me if you first predicted the predictor step and then iteratively multiple different corrector steps based on H.**
>
> To clarify, the output in Line 3 is a termination criterion (convergence tolerance $\epsilon_n$ or maximum iterations $i_n^{max}$) that controls the loop in Lines 6-8. The actual correction update in Line 7 relies on problem-specific local solver, rather than the NN output in Line 3. We have provided these problem-specific details in the Appendix A due to space constraints. We apologize for any confusion caused by this abstraction in the main text.
>
> > **W7: You should mention in the main paper that the functions you test on are 2d in Section 5.3.**
>
> We thank the reviewer for this suggestion and have explicitly stated this in the revised manuscript (Line 361).
> We have also carefully reviewed all phrasing and formulas.

---

> > ### Comment · Reviewer_pEeU · 2025-11-21
> >
> > Thank you for your very detailed rebuttal response. I appreciate that you additionally stress the unifying perspective in the paper now. Furthermore, I believe that I have better understood the method regarding the NN outputs and the overall algorithm. Additionally I thank you for clarifying details about the implementation such as reward weighting and KSD computation.
> >
> > In my opinion the strengths and weaknesses of the work remain the same. The paper is well-written and shows competitive results. At the same time, I agree with reviewer 7Gmc that it does not propose a fully new algorithm or propose new theoretical insights. Further, where prior work used "handcrafted heuristics", the presented approach uses handcrafted rewards, which I think is not strictly better.
> >
> > Overall I still think this paper is a valuable contribution because it seems to provide a relatively solid recipe for training neural-based PC approaches that seems to generalize across different types of problems. Since the authors promise to release their code in Section 8, I still lean towards accept, but would not argue with more confident reviewers about the value of the work. Hence I keep my "weak accept" score.

---

> > > ### Author Response · Authors · 2025-11-23
> > > **Official Comment by Authors**
> > >
> > > Thank you for your thoughtful feedback.
> > > We are delighted to hear that our reply addressed your concerns and appreciate your recognition of our work.
> > > In response to Reviewer 7Gmc, we clarify that our contribution lies in introducing an RL-based learning algorithm and offers valuable insights for the homotopy schedule.
> > >
> > > > To our knowledge, NPC is the first unified RL framework that leverages the homotopy-specific PC structure to find the optimal homotopy schedule for diverse tasks, which is not addressed in previous RL-driven optimization/sampling work.
> > >
> > > > A key insight of our work is that the homotopy schedule is a dominant factor governing both efficiency and stability in PC-based solvers, yet existing methods rely almost exclusively on handcrafted, task-specific heuristics.
> > >
> > > On the comment about handcrafted rewards, these offer the necessary guidance for accuracy and efficiency in training. In practice, the learned policy is not sensitive to reward weighting as shown in the reward scaling comparisons (Appendix D Tab.7). Furthermore, the RL algorithms, driven by handcrafted rewards, can automatically find schedules better than handcrafted heuristics.
> > >
> > > We again sincerely appreciate the time and effort you have devoted to reviewing our work.

---

> ### Author Response · Authors · 2025-11-19
> **Official Response by Authors -- Part 2**
>
> > **Q1: How do you weigh the two terms in your reward formulation? A specific equation would be very helpful.**
>
> The reward function comprises two components, where the Terminal Efficiency Bonus is only awarded at the termination of an episode. Consequently, the cumulative reward $R$ for an episode is formulated as: $R=(\sum_{t=1}^T\lambda_1r_t^{\text{acc}}) + \lambda_2r^{\text{eff}}$, where $r_t^{\text{acc}}$ represents the step-wise accuracy reward, $r^{\text{eff}}$ denotes the terminal efficiency bonus, and $\lambda_1, \lambda_2$ are scaling coefficients. We also included this specific reward equation in the revised manuscript (Line 287).
>
> > **Q2: For how long did you train your model? I am not familiar with the defaults of stable baselines and the info is not listed in the appendix.**
>
> The model trains for approximately 30 minutes to 1.5 hours, depending on the task.
>
> > **Q3: How did you choose your kernel for computing the KSD in the evaluation?**
>
> We use the standard RBF kernel for computing KSD, following common practice in prior work. We have added this detail to the revised appendix (Line 871).
>
> > **Q4: As you state yourself, there are prior methods that use learning in the context of PC, but which seem to not generalize as well as yours as you point out in line 111. It would be interesting to see a comparison here, to understand better at which cost the generalization of your method comes, if an at all, especially as it is one of the key claimed contributions. Have you done such comparisons?**
>
> Our method generalizes across diverse tasks with no compromise in accuracy or efficiency, as shown in our experiments.
> This capability arises from exploiting the shared predictor–corrector structure across homotopy problems.
> In contrast, prior learning-based methods focus on task-specific features rather than the general PC structure.
> For example, Simulator HC [4] trained on UPnP cannot generalize beyond this task, and iDEM [5] requires separate networks for GMM and DW-4 distributions.
>
> **References**
>
> [1] Song, Yang, et al. "Score-based generative modeling through stochastic differential equations." arXiv preprint arXiv:2011.13456 (2020).
>
> [2] Lin, Xi, et al. "Continuation path learning for homotopy optimization." International Conference on Machine Learning. PMLR, 2023.
>
> [3] Xu, Chen. "Global Optimization with a Power-Transformed Objective and Gaussian Smoothing." Forty-second International Conference on Machine Learning.
>
> [4] Zhang, Xinyue, et al. "Simulator HC: Regression-based Online Simulation of Starting Problem-Solution Pairs for Homotopy Continuation in Geometric Vision." Proceedings of the Computer Vision and Pattern Recognition Conference. 2025.
>
> [5] Akhound-Sadegh, Tara, et al. "Iterated denoising energy matching for sampling from boltzmann densities." arXiv preprint arXiv:2402.06121 (2024).

---

### Official Review · Reviewer_7Gmc · 2025-11-01

**Soundness:** 3
**Presentation:** 3
**Contribution:** 3
**Rating:** 4
**Confidence:** 4

**Summary:**

This paper proposes a Neural Predictor-Corrector (NPC) framework that leverages reinforcement learning (RL) to address homotopy problems across diverse domains, including robust optimization, global optimization, polynomial root-finding, and sampling. The core idea is to unify these traditionally independent homotopy tasks under a shared predictor-corrector (PC) structure, replacing hand-crafted heuristics for step-size selection and iteration termination with RL-learned adaptive policies. The authors employ an amortized training regime to enable one-time offline training and deployment on unseen instances, and validate NPC through experiments showing improved efficiency and stability compared to classical baselines.

**Strengths:**

1. Unified Framework for Diverse Homotopy Tasks: The paper identifies and formalizes the common PC structure underlying homotopy problems in optimization, root-finding, and sampling—an insight that helps consolidate fragmented research in these domains and highlights potential generalizability across tasks.
2. Empirical Validation Across Domains: The authors conduct comprehensive experiments on four representative homotopy tasks (Graduated Non-Convexity, Gaussian Homotopy, Homotopy Continuation, Annealed Langevin Dynamics) and provide detailed ablation studies (e.g., RL state component analysis) to support the effectiveness of NPC in improving efficiency while preserving solution accuracy.
3. Amortized Training for Practical Deployment: The amortized training design addresses a key limitation of task-specific learning methods by enabling deployment on unseen instances without per-task fine-tuning, which enhances the practical utility of the framework for real-world applications.

**Weaknesses:**

1. Limited Novelty in RL for Optimization/Sampling: The core premise of applying RL to improve optimization or sampling workflows is not new. As noted in the paper’s related work, prior studies (e.g., Li, 2019; Belder et al., 2023; Ye et al., 2025) have already explored RL for adaptive parameter tuning, optimizer design, and schedule prediction in similar problem spaces. The paper does not sufficiently distinguish NPC from these existing RL-driven optimization/sampling frameworks beyond its focus on homotopy-specific PC structures.
2. Incremental Improvement Over Traditional Methods: NPC largely builds on the well-established PC algorithm for homotopy problems and only replaces heuristic step-size/termination rules with RL policies—this constitutes a relatively minor modification rather than a paradigm shift. The framework does not introduce new theoretical insights into homotopy methods or RL for sequential decision-making; instead, it refines existing components with incremental adjustments, limiting its contribution to methodological advancement.
3. Dependence on Manual Reward Scaling: A critical practical limitation is the need for manual tuning of reward scales for each problem instance (detailed in Appendix A), which undermines the framework’s claim of being a “general solver.” This manual step not only increases the barrier to deployment but also contrasts with the goal of automating heuristic-driven decisions—an issue that the paper acknowledges but does not meaningfully address beyond proposing future work.

**Questions:**

1. Generalization to Non-Homotopy PC Tasks: The paper emphasizes unification across homotopy problems, but many non-homotopy tasks (e.g., iterative convex optimization, SGD with adaptive learning rates) also use PC-like structures. Does NPC’s RL policy generalize to these non-homotopy PC tasks, or is it inherently tied to the homotopy interpolation paradigm? If not, what limits its generalizability?
2. Comparison to Learning-Based PC Baselines: The paper compares NPC to classical PC methods but only briefly mentions learning-based baselines (e.g., Simulator HC for polynomial root-finding). Could the authors include a more detailed comparison to these learning-based alternatives?

---

> ### Author Response · Authors · 2025-11-19
> **Official Response by Authors -- Part 1**
>
> We appreciate the time and effort the reviewer dedicated to reviewing our work. We address the comments and concerns below and clarify the identified issues.
> > **W1: Limited Novelty in RL for Optimization/Sampling**
>
> Prior RL-based approaches indeed focus on task-specific adaptive schedules or optimizers.
> These methods design task-dependent state, action, reward, and neural network, making them not transferable across optimization, sampling, or algebraic tasks.
> In contrast, we show that a broad family of problems, including robust optimization, global optimization, polynomial root-finding, and sampling, share a universal predictor–corrector (PC) structure under the homotopy paradigm.
> This allows us to develop a single RL formulation (shared state, action, and neural network) that generalizes across diverse homotopy problems without redesign.
> To our knowledge, NPC is the first unified RL framework that leverages the homotopy-specific PC structure to find the optimal homotopy schedule for diverse tasks, which is not addressed in previous RL-driven optimization/sampling work.
>
> > **W2: Incremental Improvement Over Traditional Methods**
>
> We clarify that our contribution goes beyond a minor refinement of existing PC heuristics.
> A key insight of our work is that the homotopy schedule is a dominant factor governing both efficiency and stability in PC-based solvers, yet existing methods rely almost exclusively on handcrafted, task-specific heuristics.
> Our framework provides two main advances.
> First, we introduce a unified RL formulation that jointly learns adaptive predictor step sizes and corrector tolerances across diverse tasks, replacing the manually engineered schedules.
> Second, the learned policy can be applied efficiently to unseen instances within the same problem class without retraining, a capability lacking in traditional PC heuristics.
> Empirically, learning this schedule alone yields significant acceleration while preserving solution quality.
>
> > **W3: Dependence on Manual Reward Scaling**
>
> Although the NPC method requires manual reward scaling, it is not sensitive to the specific scale parameters, as shown in the following table. Specifically, we generally determine the reward scale by roughly matching the orders of magnitude of the two reward components. Once determined, this scale remains fixed for all following experiments.
>
> | Method | Reward Scaling | Convergence Steps (Training) | $\log(E_R) \downarrow$ | $\log(E_t) \downarrow$ | Iter |
> | :--- | :--- | :---: | :---: | :---: | :---: |
> | Ours+GNC | $\lambda_1 = 10^3, \lambda_2 = 10^{-3} (*)$ | 3M | -1.11 | -2.86 | 86 |
> | | $\lambda_1 = 10^2, \lambda_2 = 10^{-3}$ | 2M | -1.08 | -2.67 | 70 |
> | | $\lambda_1 = 10^3, \lambda_2 = 10^{-4}$ | 6M | -1.08 | -2.91 | 74 |
> | | $\lambda_1 = 10^2, \lambda_2 = 10^{-2}$ | Fail | - | - | - |
> | Classic GNC | - | - | -1.12 | -2.89 | 486 |
> | IRLS GNC | - | - | -1.10 | -2.90 | 141 |
>
> > $(*)$: The settings used in the paper.

---

> ### Author Response · Authors · 2025-11-19
> **Official Response by Authors -- Part 2**
>
> > **Q1: Generalization to Non-Homotopy PC Tasks**
>
> NPC is designed specifically for homotopy-based PC tasks, and its policy is not intended to generalize to non-homotopy iterative methods such as SGD or standard convex optimization.
> These methods differ fundamentally in structure: in homotopy PC, each corrector step solves a distinct optimization sub-problem along a continuously evolving target, whereas iterative methods apply incremental updates to a fixed objective.
> Because NPC explicitly models this homotopy-driven evolution, its policy does not transfer to non-homotopy settings, which fall outside the scope of this work.
>
> > **Q2: Comparison to Learning-Based PC Baselines**
>
> We have included more learning-based baselines in revised manuscript (Table 3) and we have reproduced it below for your convenience. We have to emphasized that, existing learning-based PC methods such as CPL[1], IDEM[2], and Simulator HC[3] are per-instance approaches: they train a separate model for each problem instance and do not generalize to new inputs or to other homotopy tasks. This setting is fundamentally different from NPC, whose main contribution is a unified RL training framework across robust optimization, global optimization, polynomial solving, and sampling.
> For the tasks where learning-based baselines are applicable, we already provide comparisons (e.g., Simulator HC for HC, IDEM for ALD).
> However, these methods cannot be extended to the broader set of tasks we consider, and classical PC methods remain the standard and widely accepted baselines.
> Thus, our comparisons are both appropriate and sufficient to demonstrate the advantages of NPC.
>
> | Problems | Method | $f(x^*) \downarrow$ | Iter | Time |
> | :--- | :--- | :---: | :---: | :---: |
> | **2d Ackley** | Classic GH | 0.07 | 501 | 16.25 |
> | | SLGH$_r$ | 0.12 | 1839 | 56.71 |
> | | SLGH$_d$ | 0.26 | 568 | 28.45 |
> | | PGS | 0.07 | 200 | 14.32 |
> | | CPL | 0.01 | - | 1701.61 |
> | | Ours$^2$+GH | 0.05 | 359 | 12.31 |
> | **Himmelblau** | Classic GH | 0.00 | 501 | 11.39 |
> | | SLGH$_r$ | 0.00 | 1839 | 41.70 |
> | | SLGH$_d$ | 2.57 | 75 | 2.57 |
> | | PGS | 1.18 | 200 | 11.33 |
> | | CPL | 0.00 | - | 2160.17 |
> | | Ours$^2$+GH | 0.00 | 345 | 8.91 |
> | **Rastrigin** | Classic GH | 0.00 | 501 | 23.76 |
> | | SLGH$_r$ | 0.00 | 1839 | 78.21 |
> | | SLGH$_d$ | 0.34 | 319 | 19.64 |
> | | PGS | 0.14 | 200 | 11.94 |
> | | CPL | 0.57 | - | 790.38 |
> | | Ours$^2$+GH | 0.00 | 247 | 11.84 |
>
> **References**
>
> [1] Lin, Xi, et al. "Continuation path learning for homotopy optimization." International Conference on Machine Learning. PMLR, 2023.
>
> [2] Akhound-Sadegh, Tara, et al. "Iterated denoising energy matching for sampling from boltzmann densities." arXiv preprint arXiv:2402.06121 (2024).
>
> [3] Zhang, Xinyue, et al. "Simulator HC: Regression-based Online Simulation of Starting Problem-Solution Pairs for Homotopy Continuation in Geometric Vision." Proceedings of the Computer Vision and Pattern Recognition Conference. 2025.

---

> ### Author Response · Authors · 2025-11-28
> **Official Comment by Authors**
>
> Dear Reviewer 7Gmc,
>
> I hope this message finds you well. As the discussion period is nearing its end with less than five days remaining, I wanted to ensure we have addressed all your concerns satisfactorily. If there are any additional points or feedback you'd like us to consider, please let us know. Your insights are invaluable to us, and were eager to address any remaining issues to improve our work.
>
> Thank you for your time and effort in reviewing our paper.
>
> Best regards, Authors.

---

### Author Response · Authors · 2025-11-19
**General Response**

We are sincerely grateful to the reviewers for their insightful comments and valuable suggestions. In response, we have carefully revised the manuscript. The key changes are as follows:

1. Add more experiments:
    - Add a learning-based baseline, CPL (7Gmc-Q2; Sec 5.3 Tab.3).
    - Add a comparison of different reward scaling (7Gmc-W3, CRzY-Q2; Appendix D Tab.7).
    - Add high-dimensional experiments on non-convex function minimization benchmarks (pEeU-W4; Appendix E Tab.9).
    - Add supplementary box plots for selected experiments (CRzY-W1; Appendix E Fig.5).
2. Add more discussion and details:
    - Include the explicit equation for the reward function (pEeU-Q1; Sec 4.2).
    - Clarify that results represent the average over independent runs (CRzY-W1; Sec 5.1).
    - Expand the discussion on the ablation study results (CRzY-Q3; Sec 5.6).
    - Specify the kernel used for KSD computation (pEeU-Q3; Appendix A.4.3).
    - Emphasize that the unifying perspective is the core contribution (pEeU-W1; Abstract, Sec 1), clarify the term "homotopy paradigm" (CRzY-Q1; Sec 1), and supplement the related work (pEeU-W1; Sec 2, Appendix C).

Thanks again for all the effort and time, and we look forward to further discussions if there are any more questions.

---

### Meta-Review · Area_Chair_8Qze · 2026-01-13

**Summary:**

This paper proposes a unified RL approach for learning the configuration in the predictor-corrector structure, replacing hand-crafted heuristics for step-size selection and iteration termination with RL-learned adaptive policies. The neural predictor-corrector (NPC) through experiments demonstrated improved efficiency and stability and surpassed classical baselines.
In the rebuttal, authors clarified the novelty of the proposed approach, multiple trials, and reward scales. Some new comparative results were added, showing that the RL-based approach is better than broad learning-based baselines. There are still some concerns that were not fully addressed, such as incremental performance improvement. While the added experiment with high dimensions verified the effect of the approach, the results are limited and can be extended.
This work can be accepted since most concerns and questions are clearly addressed.

**Reviewer Concerns:**

The main concerns relate to the novelty, the extent of performance improvements, and the lack of experiments in high-dimensional settings. The authors have clarified that the proposed RL-based approach provides a unified framework aimed at improving performance across a range of different problems. As such, the novelty is primarily application-oriented rather than stemming from a fundamentally new methodology. In response, the authors have added experiments in high-dimensional settings and included additional baselines, which substantially strengthen the empirical evaluation.

**Reviewer Scores:**

During the rebuttal, one reviewer increased their score from 6 to 8. Another reviewer chose to maintain a score of 6, noting that no fundamentally new algorithm was proposed. A third reviewer assigned a score of 4, citing the same concern regarding limited novelty.

---

### Decision · Program_Chairs · 2026-01-26

Accept (Poster)